# A parameterization scheme for the floating wind farm in a coupled atmosphere-wave model (COAWST v3.7)

Shaokun Deng[1], Shengmu Yang[2], Shengli Chen[1], Daoyi Chen[1], Xuefeng Yang[1],Shanshan Cui[1]

[1]Institute for Ocean Engineering, Shenzhen International Graduate School, Tsinghua University, Shenzhen, China
[2]College of Meteorology and Oceanography, National University of Defense Technology, Changsha, China

*Correspondence to*: Shengli Chen (shenglichen@sz.tsinghua.edu.cn)

**Abstract.** Coupling the Weather Research and Forecasting (WRF) model with wind farm parameterization can be effective in examining the performance of large-scale wind farms. However, the current scheme is not suitable for floating wind turbines. In this study, a new scheme is developed for floating wind farm parameterization (FWFP) in the WRF model. The impacts of the side columns of a semi-submersible floating wind turbine on waves are first parameterized in the spectral wave model (SWAN) where the key idea is to consider both inertial and drag forces on side columns. A machine learning model is trained using results from idealized high-resolution SWAN simulations and then implemented in the WRF to form the FWFP. The difference between our new scheme and the original scheme in a realistic case is investigated using a coupled atmosphere-wave model. The results show that the original scheme has a lower power output in most of the grids with an average of 12% compared to the FWFP scheme. The upstream wind speed is slightly increased compared to the original scheme (<0.4 m/s), while the downstream wind speed is decreased, but by a much larger magnitude (<1.8 m/s). The distribution of the difference in TKE corresponds well to that of the wind speed, and the TKE budget reveals that the difference in TKE in the rotor region between the two schemes is mainly due to vertical wind shear. This demonstrates that the FWFP is necessary for both predicting the wind power and evaluating the impact of floating wind farms on the surrounding environment.

## 1 Introduction

Wind energy has shown great potential for development in recent years. The number of wind farms that have been built is enormous, and there are predictions that wind power generation will increase in the future. (Pryor et al., 2020). The pre-assessments of wind farms are not suitable to be investigated with the computational fluid dynamics (CFD) models and large-eddy simulation (LES) models due to great computational expense and feedback effects that cannot be captured by high-resolution non-meteorological microscale models alone. Furthermore, the relevant physical processes, which are important for large wind farms, are also not included in the engineering wake models (Emeis, 2010). Mesoscale models coupled with LES can theoretically take advantage of both, but most studies focus on one-way coupling rather than two-way coupling and involve high computational cost (Carvalho et al., 2013; Santoni et al., 2018; Temel et al., 2018). Currently, an

important tool for investigating large-scale wind sources and wake interferences is mesoscale model with a wind farm parameterization.

There are two different methods to parameterize the wind farm in mesoscale models: implicit and explicit methods. In implicit parameterization, it is common to modify the surface roughness to characterize the effect of wind farms. The explicit methods parameterize the wind farm effect as a momentum sink on the mean flow. Previous results have shown that explicit
methods present a more physically consistent representation of wind farm effects and result in more realistic simulations (Fitch et al., 2013; Fitch, 2015). The explicit methods also have the advantage of taking info account how wind speeds interact with the lower surface (Du et al., 2017; Vanderwende and Lundquist, 2016). Most of wind farm parameterizations are conducted in the free, open-source Weather Research and Forecasting (WRF) model, which already includes the Fitch wind farm parameterization in its release (Fitch et al., 2012). The original Fitch scheme has been the subject of a number of
recent developments and modifications. Most of studies have focused on sub-grid effects of wind turbines (Abkar and Porte-Agel, 2015; Ma et al., 2022a, b; Pan and Archer, 2018; Redfern et al., 2019), and a few studies have focused on TKE treatment in wind farm parameterization (Archer et al., 2020; Volker et al., 2015).

Global offshore wind power development is moving from near-shore to deeper waters, where floating offshore wind turbines have advantage over bottom fixed offshore wind turbines in water depths greater than 50 m (Diaz, 2020; Roddier et
al., 2010). Floating wind turbines are divided into many categories, among which the semi-submersible floating wind turbine is a popular type of floating wind turbine structure in the industry. However, the semi-submersible floating offshore wind turbines can have a substantial impact on waves due to floating platforms, which in turn leads to major changes in roughness length of ocean surface. Changes in the roughness length in turn affect the wind field through momentum transfer between the atmosphere and the waves. And the effect of the wave field on the wind field can reach up to the height of the turbine,
according to previous studies (AlSam et al., 2015; Jenkins et al., 2012; Kalvig et al., 2014; Paskyabi et al., 2014; Porchetta et al., 2021; Wu et al., 2020; Yang et al., 2014; Zou et al., 2018). This suggests that the current wind farm parameterization is not suitable for semi-submersible floating wind farms because it does not account for the change in roughness length caused by large floating platforms.

In contrast to studies investigating the influence of offshore wind farm wake on waves, few studies have investigated
the influence of wind farm structures (piles) on waves through the effects of drag dissipation. Ponce de Leon et al. (2011) used a wave model to study the impact of an offshore wind farm on nearby waves. They represented each monopile foundation as a dry point (land) in the model. They found that the method blocked the propagation of the wave energy and caused a slight change in the direction of the wave. Alari and Raudsepp (2012) found that the impact of the wind turbine on the significant wave height (SWH) was very marginal, with changes of the SWH smaller than 1 % at areas shallower than 10
m depth. Molen et al. (2014) conducted sensitivity experiments to study the influence of turbine spacing and size of wind farm on the SWH, and found that the SWH could be reduced by up to 9.58 %. McCombs et al. (2014) evaluated the impact of an offshore wind farm on waves in Lake Ontario using a coupled wave-hydrodynamic model. In contrast to previous studies, they simulated the offshore wind farm with the application of a transmission coefficient in the wave model. The

results indicated that with changes in SWH predicted to be less than 3 %. These previous studies simulate the wind turbine in the model as a dry grid point, which has two limitations, 1) the model resolution is too high to implement for large-scale offshore wind farm scenarios, 2) it can only represent the diffraction effects, however, wave forces include drag and inertial forces (Isaacson, 1979; Morison et al., 1950). By parameterizing both the drag and inertial forces in the numerical model, the impact of the offshore wind turbine/farm on the waves can be analyzed more accurately.

In this study, a floating wind farms parameterization (FWFP) scheme is developed to in the WRF Model represent the effect of the offshore wind farm on surface waves. In Section 2, the wave energy dissipation due to the inertial forces of waves is implemented in SWAN. The model configuration and results of high-resolution idealized simulations are presented in Section 3. In Section 4, we propose a machine learning module used to fit the effect of wave inertial forcings represented in high-resolution SWAN simulations. Section 5 describes how the floating wind farm parameterization scheme is implemented in the WRF, and presents the results and the analysis of the wind speed deficit, power output, and the influence of the new scheme on the turbulent kinetic energy. The conclusion is given in Section 6.

## 2 Parameterization of the wave inertial force in SWAN

SWAN is a third-generation phase-averaged spectral wave model (Booij et al., 1999). SWAN has a function to account for wave damping over a vegetation (VEG) at variable depths. The cylinder approach proposed by Dalrymple et al. (1984) is a well-known method for expressing wave dissipation due to vegetation. In this approach, the energy loss is calculated based on the actual work done by the force of the plant on the fluid, expressed by the Morison equation. Two modifications convert the VEG module into the semi-submersible floating wind turbine module. The first modification then is that the module only needs to calculate the energy dissipation in the top layer (layer1 in Figure 1) and set $d$ (column draft depth) to a constant ($d$=20 m is used in this paper).

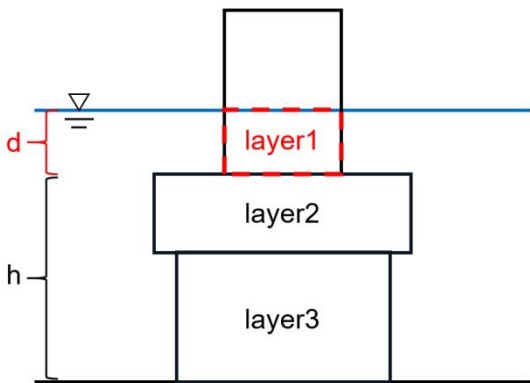

**Figure 1.** Layer schematization for vegetation in SWAN. The submerged part of the plant can be layered by diameter as well as by drag coefficient (layer1, layer2,...). Each layer has a different thickness, $d$ for layer1.

Another important modification of the VEG module concerns the energy dissipation due to wave forces. In the VEG module, the wave force is derived from the drag force in a Morison type equation with the inertial forces neglected. Since the vegetation is assumed to be a cylinder with a small diameter, the drag force is considered to be dominant. However, for the floating offshore wind turbine, the diameter of the cylinder cannot be neglected compared to the wavelength. The wave forces become more complex and require the consideration of inertial forces. The equation for the energy dissipation due to inertial forces ($F_{iner}$) can be derived from the work of Morison et al. (1950).

$$D = \int_{-(h+d)}^{-h} F_{iner} u dz = \int_{-(h+d)}^{-h} C_M \cdot \rho V \cdot \frac{\partial u}{\partial t} \cdot u dz = \int_{-(h+d)}^{-h} \rho C_M \frac{\pi}{4} b^2 \frac{\partial u}{\partial t} u dz, \tag{1}$$

where $\rho$ is the fluid density, $V$ is the volume, $C_M$ is the inertial force coefficient, $b$ is the cylinder diameter, $h+d$ is the water depth, $d$ is the draft depth (Figure 1), $u$ is the horizontal fluid velocity. Following Kobayashi et al. (1993), the fluid convective accelerations and stresses are assumed to be negligible in this region as a first approximation. The linearized horizontal and vertical momentum equations can be expressed as

$$\rho \frac{\partial u}{\partial t} = -\frac{\partial p}{\partial x}, \tag{2}$$

$$\rho \frac{\partial w}{\partial t} = -\frac{\partial p}{\partial z}, \tag{3}$$

The Eqs. (2) and (3) are used to express $u$ and $w$ (vertical fluid velocity) in terms of $p$ (the dynamic pressure).

$$u = \frac{k_c}{\rho \omega} p, \tag{4}$$

$$w = -\frac{i}{\rho \omega} \frac{\partial p}{\partial z}, \tag{5}$$

To solve the linearized problem, it is convenient to introduce the complex wave number $k_c = k + ik_i$, $k_i$ is the exponential decay coefficient, $k$ is the wave number.

The continuity equation is given by

$$\frac{\partial u}{\partial x} + \frac{\partial w}{\partial z} = 0, \tag{6}$$

Substituting (4) and (5) into (6) and solving the resulting equation with the conditions $w = 0$ at $z = -(h+d)$, $p$ is given by

$$p = \rho g \frac{H}{2} \left[ \cosh(k_c h) - \frac{\omega^2}{gk} \sinh(k_c h) \right] \frac{\cosh[\alpha(z+h+d)]}{\cosh(\alpha d)} exp[i(k_c x - \omega t)], \tag{7}$$

with $\alpha \cong k_c(1 - i\varepsilon)$, $\omega$ is the wave angular frequency, $H$ is the wave height, $\varepsilon$ is the dimensionless damping coefficient.

For the case of weak damping in which dimensionless damping coefficient $\varepsilon$, Eq, (7) with $k_c = k(1 + i\delta)$ and $\alpha \cong k[1 + i(\delta - \varepsilon)]$ can be shown to be approximated as

$$p \cong \rho g \frac{H}{2} \frac{\cosh[k(h+d+z)]}{\cosh[k(h+d)]} [1 + i\delta c_2 + i(\delta - \varepsilon)c_3] exp[i(kx - \omega t)] , \qquad (8)$$

with

$$c_2 = \frac{\sinh kh \sinh[k(h+d)] - kh \sinh kd}{\cosh kd}, \qquad (9)$$

$$c_3 = k(h+d+z)\tanh[k(h+d+z)] - kd \tanh kd, \qquad (10)$$

$$\varepsilon = \frac{C_D bH}{9\pi} \frac{\sinh 3kd + 9 \sinh kd}{(2kd + \sinh 2kd)\sinh[k(h+d)]}, \qquad (11)$$

$$\delta = \frac{k_i}{k} \cong \varepsilon \frac{2kd + \sinh 2kd}{2k(h+d) + \sinh[2k(h+d)]}, \qquad (12)$$

Substitution of the real parts of Eq. (8) into Eq. (2) yields

$$u = \frac{gkH}{2\omega} \frac{\cosh[k(h+d+z)]}{\cosh[k(h+d)]}, \qquad (13)$$

$$\frac{\partial u}{\partial t} = \frac{k}{\rho\omega} \frac{dp}{dt} = \frac{gkH}{2} \frac{\cosh[k(h+d+z)]}{\cosh[k(h+d)]} [\delta c_2 + (\delta - \varepsilon)c_3], \qquad (14)$$

Substitution of Eqs. (13), (14) and (10) into Eq. (1) yields

$$D = \frac{\rho C_M \pi b^2 k(gH)^2}{16\omega[\cosh k(h+d)]^2} \left[ \delta c_2 \frac{\sinh 2kd + 2kd}{4} + (\delta - \varepsilon) \frac{2kd \cosh 2kd - \sinh 2kd - kd \tanh kd(\sinh 2kd + 2kd)}{8} \right], \qquad (15)$$

In order to simplify the derivation of the equation, $\varepsilon = \frac{C_D bH}{9\pi} c_5$ and $\delta = \varepsilon c_4$ are substituted into Eq. (15)

$$D = \frac{\rho C_M C_D b^3 k g^2 H^3}{144\omega[\cosh k(h+d)]^2} \left[ c_2 c_4 c_5 \frac{\sinh 2kd + 2kd}{4} + c_5(c_4 - 1) \frac{2kd \cosh 2kd - \sinh 2kd - kd \tanh kd(\sinh 2kd + 2kd)}{8} \right], \qquad (16)$$

Waves can be described by a joint distribution of wave height, period (or frequency), and direction. For simplicity of analysis, it is usually assumed that all wave heights are related to an average peak period and mean direction. The model developed by Mendez and Losada (2004) describes the transformation of the wave height distribution assuming an unmodified Rayleigh distribution, then the average energy dissipation is as follow,

$$\langle D \rangle = \frac{\rho C_M C_D b^3 k_p g^2}{144\omega_p[\cosh k_p(h+d)]^2} \left[ c_2 c_4 c_5 \frac{\sinh 2k_p d + 2k_p d}{4} + c_5(c_4 - 1) \frac{2k_p d \cosh 2k_p d - \sinh 2k_p d - k_p d \tanh k_p d(\sinh 2k_p d + 2k_p d)}{8} \right] \cdot \int_0^\infty H^3 p(H) dH \ , \qquad (17)$$

$$\int_0^\infty H^3 p(H) dH = \frac{3\sqrt{\pi}}{4} H_{rms}^3, \qquad (18)$$

where the terms with subscript p are associated with the peak period (the subscript p is neglected in the following). And $p(H)$ is the Rayleigh probability density function, $H_{rms}$ is the root-mean-square wave height. Substitution of Eqs. (18) into Eq. (17) and dividing by the bulk density of the fluid yields

$$\langle D \rangle = \frac{\rho C_M C_D b^3 gk}{144\omega[\cosh k(h+d)]^2} \frac{3\sqrt{\pi}}{4} H_{rms}^3 \left[ c_2 c_4 c_5 \frac{\sinh 2kd + 2kd}{4} + c_5(c_4 - 1) \frac{2kd \cosh 2kd - \sinh 2kd - kd \tanh kd(\sinh 2kd + 2kd)}{8} \right], \quad (19)$$

The root-mean-square wave height and the total wave energy have such a relationship, $H_{rms}^2 = 8E_{tot}$, and substituting it into Eq. (19) yields,

$$\langle D \rangle = \frac{1}{8} \sqrt{\frac{\pi}{72}} \frac{C_M C_D b^3 gk}{\omega[\cosh k(h+d)]^2} [2c_2 c_4 c_5 (\sinh 2kd + 2kd) + c_5(c_4 - 1)[2kd \cosh 2kd - \sinh 2kd - kd \tanh kd(\sinh 2kd +$$
$$2kd)]]E_{tot}^{3/2}, \quad (20)$$

Eq. (20) is the equation calculating the energy dissipation due to the inertial force. In the study, the magnitudes of the inertial force and the drag force are calculated and compared for the cylinders with diameters of 10 m and 1 m.

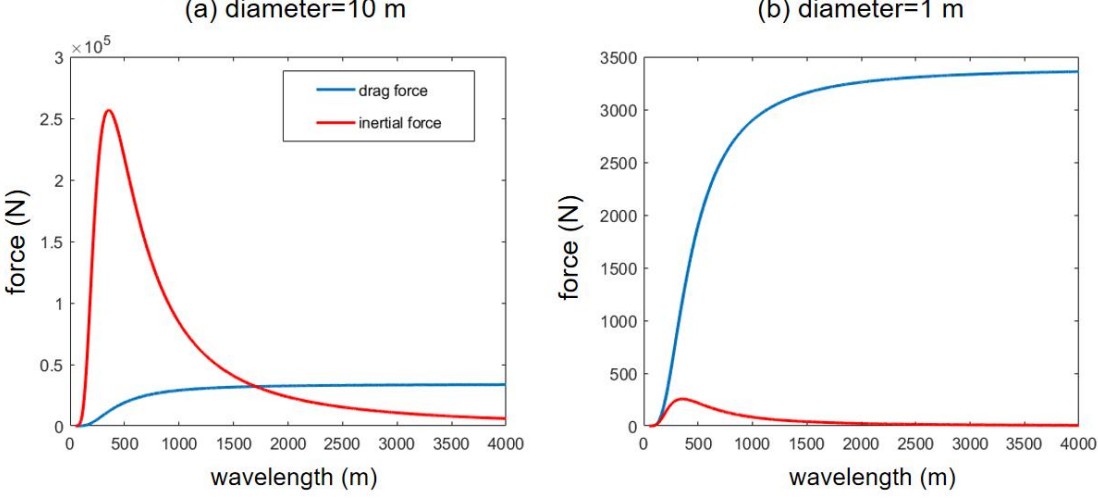

**Figure 2.** Inertial forces (red solid line) and drag forces (blue solid line) for cylindrical diameters of (a) 10 m and (b) 1 m (incident wave height of 3 m, drag coefficient of 1.2, draft depth of 20 m, water depth of 80 m)

The case is set with the incident SWH of 3 m, the drag coefficient of 1.2, the draft depth of 20 m, and the water depth being 80 m. When the cylinder diameter is 10 m (Figure 2a), the average wavelength of the incident wave is within 1700 m, which makes the inertial force larger than the drag force. However, when the cylinder diameter is 1 m (Figure 2b), the inertial force is always smaller than the drag force. As the wavelength increases (the scale becomes smaller), the drag force becomes larger relative to the inertial force, which is consistent with the assumption of the VEG module that the inertial force could be neglected, but the inertial force cannot be ignored for the side column of the floating offshore wind turbine. Therefore, the VEG module in SWAN is modified to include the inertial force to be applicable for the floating wind turbine.

## 3 Idealized high-resolution simulations

As shown in Section 2, the floating offshore wind turbine module is developed for SWAN, and its impact on waves is examined using high-resolution numerical experiments in this section.

The rectangular domain of the idealized high-resolution experiments is shown in Figure 3, with $100 \times 200$ cells, a horizontal resolution corresponding to the column diameter of 10 m, and a water depth of 50 m. The position of the column is at the center of the computational domain. The incident SWH is 3 m, the mean wave period is 12 s, propagating from east to west, and the shape of the spectra is from the JONSWAP spectrum. Because of the small computational domain, the model uses stationary computation which converges after several time steps.

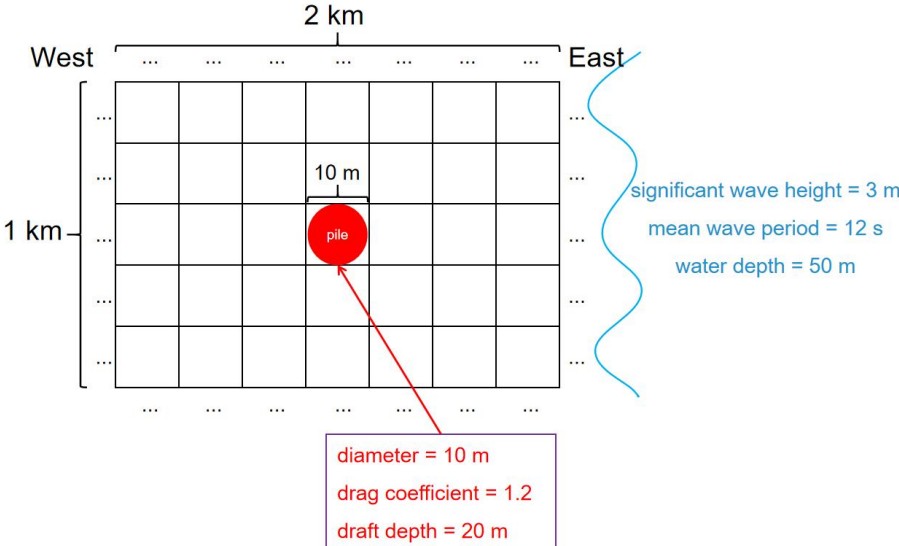

**Figure 3.** Experimental design of high-resolution idealized simulations, the blue curve on the right side represents waves (SWH=3 m, mean wave period=12 s, and water depth=50 m).

Two experiments are conducted, one to study the influence of the column on the waves caused by the drag force only (ExpDragS), and the other to examine the influence caused by both the drag force and the inertial force (ExpInerS). It can be noted that when the energy dissipation is caused by the drag force only, the SWH attenuation is only ~0.2 m (Figure 4a), and the "wake" phenomenon occurs in the wave field. The angle of the mean wave direction is shifted by about 1° around the column and the horizontal distribution is symmetric along the axis y=0 (Figure 4d). The mean wave length is increased by about 10 m (Figure 4g). When the inertial forces are taken into account, the energy dissipation is larger, which makes the SWH attenuation more significant, which is about 1.4 m (Figure 4b), indicating an attenuation of 50 % SWH. The mean wave direction deviation around the column is also relatively large, reaching about 5° (Figure 4e), and the mean wave length is about 24 m longer (Figure 4i) than that of ExpDragS.

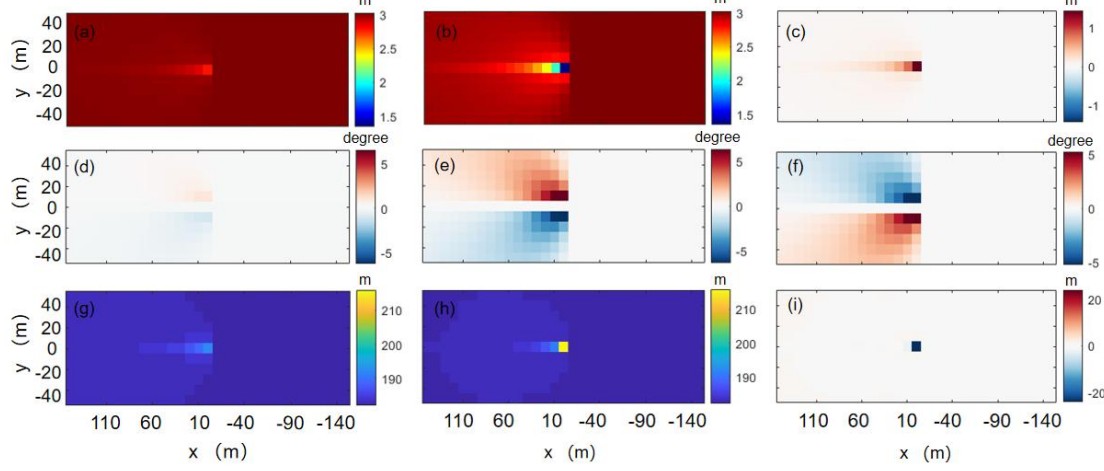

**Figure 4.** Significant wave height of (a) ExpDragS and (b) ExpInerS, (c) difference in significant wave height, (d) mean wave direction deviation of ExpDragS and (e) ExpInerS, (f) difference in mean wave direction deviation, (g) mean wave length of ExpDragS, and (h) ExpInerS, (i) difference in mean wave length

## 4 Machine learning parameterization

The results of the idealized high-resolution SWAN simulations in Section 3 show the impact of the floating offshore wind turbine's side columns on the waves, including the SWH attenuation, symmetric changes in mean wave direction, and an increase in the mean wave length. However, it is computationally expensive to run a ~10 m resolution SWAN model.

Using Machine Learning (ML) to better parameterize unresolved processes in mesoscale and climate models has received much attention in recent years (O'Gorman and Dwyer, 2018; Gettelman et al., 2020; Seifert and Rasp, 2020). With the rise of scientific ML and its widespread use in the geosciences, the design of parametrizations using ML algorithms has become a trend in model development. To build an appropriate model, a large amount of data is needed for training. Nevertheless, the observational data on the impact of the floating offshore wind turbine on waves are scarce. As a result, the outputs of the high-resolution SWAN simulations in Section 3 are employed to train the ML model.

From the equations in Section 2, we can note that when the inertial force coefficient, drag force coefficient, and cylindrical diameter are determined, the energy dissipation caused by the wave force is only related to the water depth, incident SWH, and mean wave period (or peak period). We design a series of ideal experiments with different water depths, incident SWH, and mean wave periods. The SWH is taken from 2 m to 4 m with 0.1 m interval. The peak wave period is taken from 7.4 s to 7.6 s, 8.4 s to 8.6 s, 9.6 s to 9.8 s, 11.0 s to 11.2 s with an interval of 0.1 s, and the water depth is selected from 53 m to 98 m with an interval of 5 m. This has a total number of 2520 (21 × 12 × 10) experimental groups. We then select simulated data that do not include water depths of 58 m, 78 m, 98 m to train several machine learning (regression) models, since data from these three water depths would be used as validation data. The inputs are incident SWH, water depth, and peak wave period, and the output is SWH after energy dissipation. These models can be classified into four main

categories: linear regression models, regression tree models, support vector machines (SVM), and Gaussian process regression (GPR). These four categories of ML models are described in detail in the Appendix.

As shown in Figure 5, after training, the GPR model with the Matern 5/2 kernel (covariance) function fits best with a minimum root mean square error (RMSE) of 0.0033 m (Figure 5d). It may be that the advantage of GPR is mainly in dealing

with nonlinear and small sample data. The validation data are also used to analyze the strengths and weaknesses of these four ML models and to prevent overfitting when training the models. Figure 6 shows that the Matern 5/2 GPR model still performs best and the Stepwise Linear Regression model still performs worst. On the other hand, the medium SVM model does not necessarily outperform the fine tree model in some cases. The ML model can be coupled with CFD, LES models and mesoscale meteorological models to predict the effect of the floating offshore wind turbine side columns on waves

without the need for high-resolution SWAN simulations.

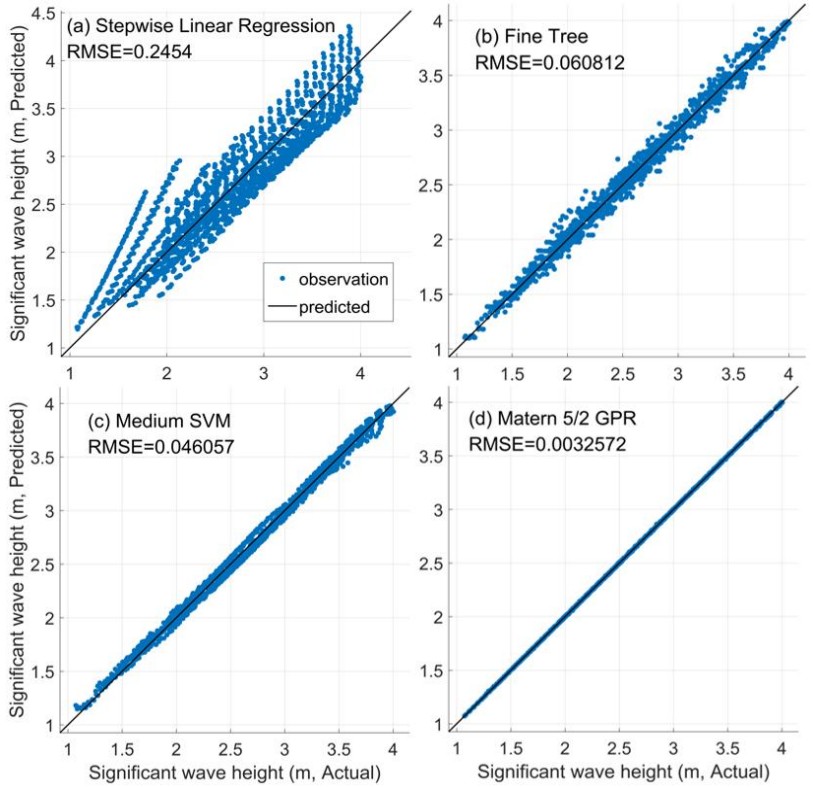

**Figure 5.** Significant wave height scatter plot of the comparison between training data and four typical ML regression models: (a) Stepwise linear regression (b) Fine tree (c) Medium SVM (d) Matern 5/2 GPR.

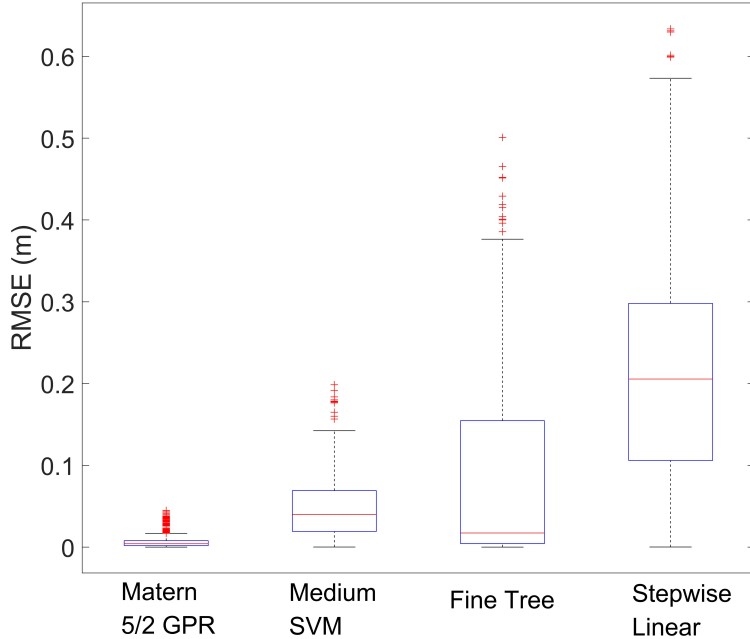

**Figure 6.** Boxplots of RMSE for four typical ML regression models using validation data. The boxplots show the median (horizontal line), 25th to 75th percentile (box) and 5th to 95th percentile (whiskers). The whiskers extend to the most extreme data points not considered outliers, and the outliers are plotted individually using the ' + ' marker symbol.

## 5 Parameterization in the WRF model

### 5.1 Implementation of parameterization in the WRF model

The rate of kinetic energy loss in the grid cell in the original wind farm parameterization scheme (Fitch et al., 2012) is equal to the kinetic energy loss due to the wind turbine in the grid,

$$-\frac{1}{2}N_{ij}\Delta x \Delta y \rho C_T V_{ijk}^3 A_{ijk} = \Delta x \Delta y (z_{k+1} - z_k)\rho V_{ijk}\frac{\partial V_{ijk}}{\partial t}, \tag{21}$$

where $V_{ijk}$, is the horizontal wind speed, $N_{ij}$ is the number of turbines per square meter, $\rho$ is the air density, $C_T$ is the thrust coefficient of a wind turbine, $\Delta x, \Delta y$, are the horizontal grid size in the zonal and meridional directions respectively, $z_k$ is the height at model level $k$, $A_{ijk}$ is the cross-sectional rotor area of one wind turbine bounded by model levels $k, k+1$ in grid cell $i, j$.

For a semi-submersible floating wind turbine, the SWH around the turbine is considerably affected. As a result, the roughness of ocean surface nearby is also changed. This means that the inflow wind speed on the left side of Eq. (21) is no longer the horizontal wind speed in the grid cell $i, j$, and the two must be distinguished.

$$-\frac{1}{2}N_{ij}\Delta x \Delta y \rho C_T V_{ijk|wt}^3 A_{ijk} = \Delta x \Delta y (z_{k+1} - z_k)\rho V_{ijk}\frac{\partial V_{ijk}}{\partial t}, \tag{22}$$

where $V_{ijk|wt}$ is the recalculated inflow wind speed at the wind turbine site, so a new equation for the momentum tendency term is given as

$$\frac{\partial V_{ijk}}{\partial t} = -\frac{N_{ij}C_T V_{ijk|wt}^3 A_{ijk}}{2(z_{k+1}-z_k)V_{ijk}}, \tag{23}$$

The corresponding component forms require modification as well


$$\frac{\partial u_{ijk}}{\partial t} = \frac{u_{ijk}}{V_{ijk}} \cdot \frac{\partial V_{ijk}}{\partial t} = -\frac{N_{ij}C_T V_{ijk|wt}^3 A_{ijk} u_{ijk}}{2(z_{k+1}-z_k)V_{ijk}^2}, \tag{24}$$

$$\frac{\partial v_{ijk}}{\partial t} = \frac{v_{ijk}}{V_{ijk}} \cdot \frac{\partial V_{ijk}}{\partial t} = -\frac{N_{ij}C_T V_{ijk|wt}^3 A_{ijk} v_{ijk}}{2(z_{k+1}-z_k)V_{ijk}^2}, \tag{25}$$

The tendency of turbulent kinetic energy (TKE) and the power generated by the wind turbine also need to be modified

$$\frac{\partial P_{ijk}}{\partial t} = \frac{N_{ij}C_P V_{ijk|wt}^3 A_{ijk}}{2(z_{k+1}-z_k)}, \tag{26}$$

$$\frac{\partial TKE_{ijk}}{\partial t} = \frac{N_{ij}C_{TKE} V_{ijk|wt}^3 A_{ijk}}{2(z_{k+1}-z_k)}, \tag{27}$$

where $C_P$ is the power coefficient and $C_{TKE}$ denotes the TKE coefficient calculated by $C_{TKE} = C_T - C_P$.

The variables exchanged between WRF and SWAN is shown in Figure 7. WRF provides 10-m surface wind to SWAN, whereas SWAN returns SWH, peak wave length, and peak wave period to WRF. This variable exchange is implemented in the coupled model. The trained GPR model needs water depth as the input, thus we implement SWAN to provide water depth to WRF. Specifically, we incorporate the GPR model into the surface layer parameterization module of WRF. As a

result, the SWH affected by the floating offshore wind turbine can be calculated directly in the surface layer parameterization module to obtain the roughness length, frictional velocity, and other variables. The above variables are then passed to the Planetary Boundary Layer Driver module to calculate the three-dimensional wind speed affected by the boundary layer. The three-dimensional wind speed at the wind turbine location is also passed to the new wind farm parameterization.

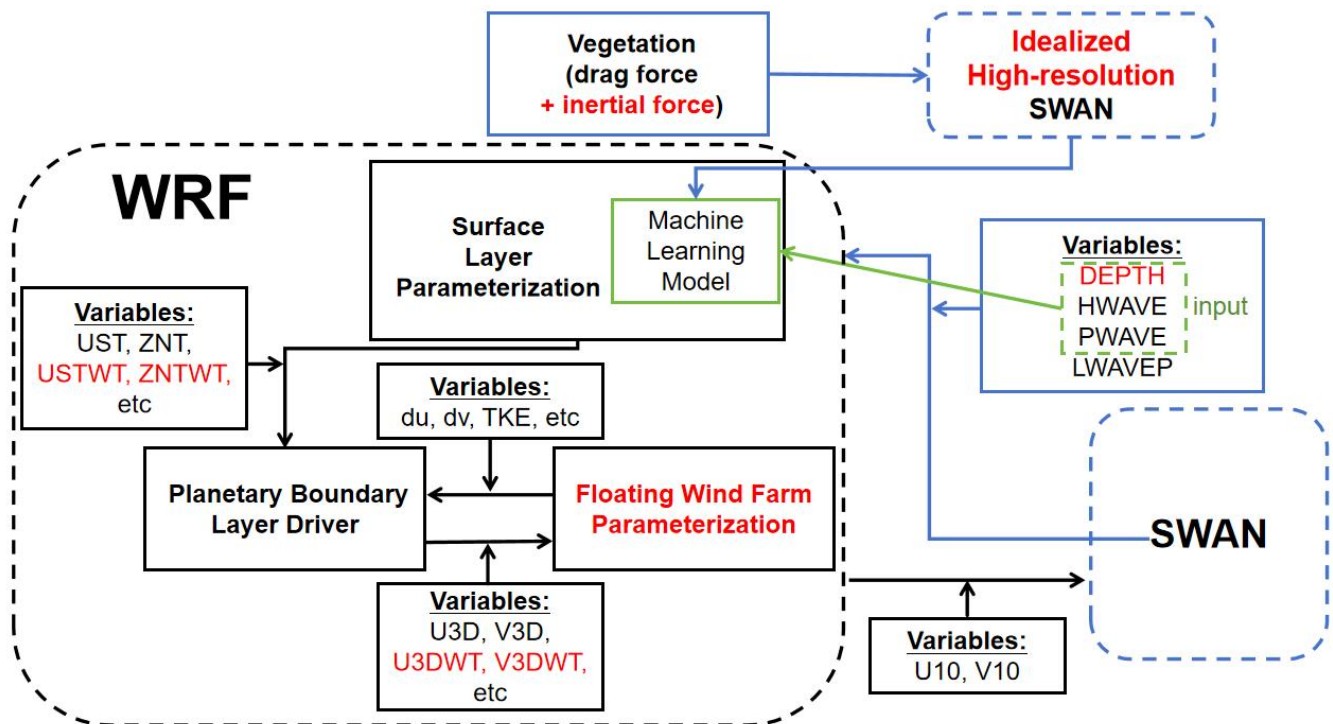

**Figure 7**. Flow chart of floating offshore wind farms parameterization implemented in the coupled model (HWAVE = significant wave height, LWAVEP = peak wave period, PWAVE = peak wave length, DEPTH= water depth, U10 = zonal wind at 10 m, V10 = meridional wind at 10 m, UST= frictional velocity, USTWT= frictional velocity at the wind turbine, ZNT = roughness length, ZNTWT = roughness length at the wind turbine,  U3D = three-dimensional zonal winds, V3D = three-dimensional meridional winds, U3DWT = three-dimensional zonal winds at the wind turbine, V3DWT = three-dimensional meridional winds at the wind turbine, TKE = turbulent kinetic

energy, du = zonal momentum increment, dv = meridional momentum increment).

### 5.2 Model configuration

The model (COAWST, Warner et al., 2010) used to run coupled simulations in this study activates only the atmospheric model (WRFv4.2.2) and the spectral wave model (SWANv41.31).

Initial and lateral boundary conditions for the WRF model are derived from the National Centers for Environmental

Prediction Global Data Assimilation System (GDAS) Final Analysis with temporal resolution of 6-hour and horizontal resolution of 0.25° (NCEP, 2015). The WRF model is configured with 47 vertical levels, where 23 levels are below 1000 m and 15 levels intersect the rotor region. The vertical spacing of the grid on the levels spanned by the wind turbine rotor is approximately 14 m. The WRF model is configured with two nested domains with horizontal resolution of 9 km and 3 km (Figure 8a). The outer domain (D01) has 300 × 220 grids and the inner domain (D02) has 250 × 163 grids. The major

physical parameterization schemes are summarized in Table 1. The wind farm is located in the northern South China Sea where the water depths range from 50 and 63 m (Figure 8a). The distance between the turbines is approximately 1 km. The

thrust and power coefficients of the LEANWIND 8 MW reference turbine (LW) are presented in Figure 8b. This floating wind turbine is rated at 8 MW, with a rotor diameter of 164 m, a hub height of 110 m, and a cut-in wind speed of 4 m/s and a cut-out wind speed of 25 m/s (Desmond et al., 2016).

SWAN uses a single domain with 8 km horizontal resolution, which is smaller than the WRF outer domain in this study (Figure 8a). The corresponding parameterization schemes are shown in Table 1. The spectrum is discretized using 24 logarithmically-spaced frequency bins from 0.04 to 1.00 Hz and 36 directional bins with 10° spacing. The boundary conditions are taken from the WaveWatch III (WW3) model (WW3DG, 2019). The nonstationary mode of SWAN is used. The WRF model is coupled with the SWAN model every 10 minutes. The simulations are integrated first for 12 hours

without the turbines to reach a steady state, and then run for another 6 hours for comparison (i.e., from 0000 UTC on 1 January to 1800 UTC on 1 January 2019). A reference simulation (control run, referred as WRF-CTL) is performed without the wind farm. Another simulation (WRF-Fitch) is conducted with the Fitch wind farm parameterization. A third simulation (WRF-FWFP) is performed with the new proposed floating wind farm parameterization.

**Table.1** Physical parameterization schemes used in coupled model.

|  | Physics process | Parameterization scheme |
|---|---|---|
| WRF | Microphysics | Single-Moment 6-class (Hong and Lim, 2006; Hong et al., 2006) |
|  | Longwave Radiation | Rapid Radiative Transfer Model (Mlawer, 1997) |
|  | Shortwave Radiation | Dudhia (Dudhia, 1989) |
|  | Surface Layer | MYNN (Nakanishi and Niino, 2009) |
|  | Land Surface | thermal diffusion (Dudhia, 1996) |
|  | Planetary Boundary Layer | Mellor-Yamada-Nakanishi-Niino 2.5-level (Nakanishi and Niino, 2009) |
|  | Cumulus | Grell-Freitas ensemble (Grell and Freitas, 2014) |
|  | Roughness | COARE-Taylor-Yelland (Taylor and Yelland, 2001) |
| SWAN | Depth-induced wave breaking | Constant (1.0, 0.73) (Battjes and Janssen, 1978) |
|  | Bottom friction | Madsen (0.05) (Madsen et al., 1988) |
|  | Wind input | Komen (Komen et al., 1984) |
|  | Whitecapping | Komen (Komen et al., 1984) |


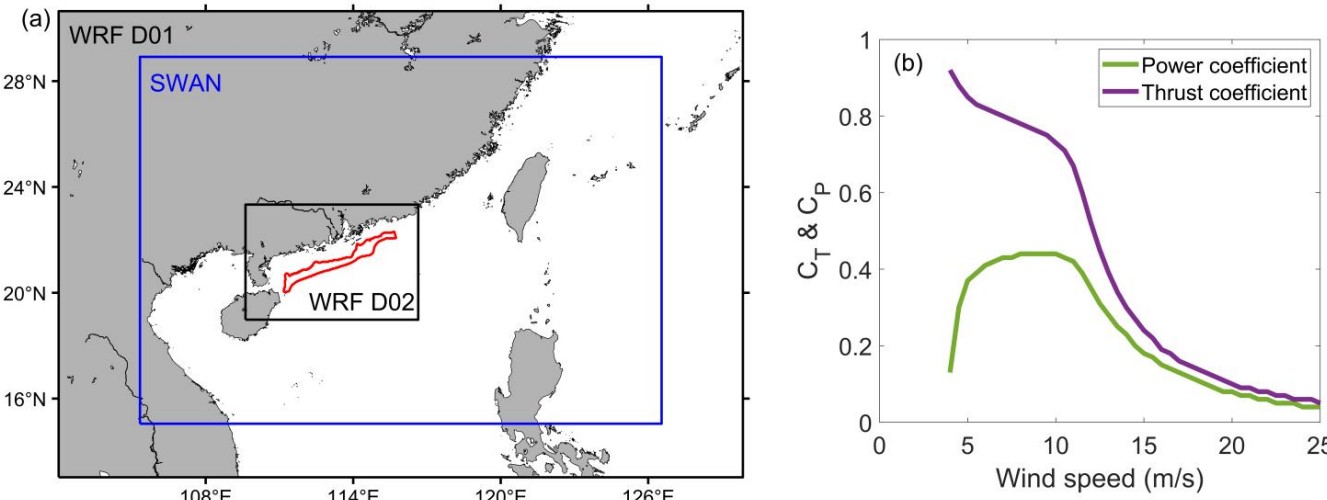

**Figure 8.** (a) The model domain used in COAWST, the red solid line indicates the outer boundary of the wind farm, (b) the thrust and power coefficients curves of the LW 8 MW wind turbine.

### 5.3 Model validation

To validate SWAN results, the simulated SWH is compared with observations of the satellite data Jason-3 (Lillibridge, 2019) (Figure 9). The model is also run for an additional 126 hours for further validation (1800 UTC 01 January to 0000 UTC 07 January). It is evident that the model generally performs well on the wave simulation for the satellite tracks (Pass 38 and Pass 88). The SWH in the model is a bit underestimated on the track Pass 12 and overestimated on the track Pass 51. Generally, the model results have a reasonable performance.

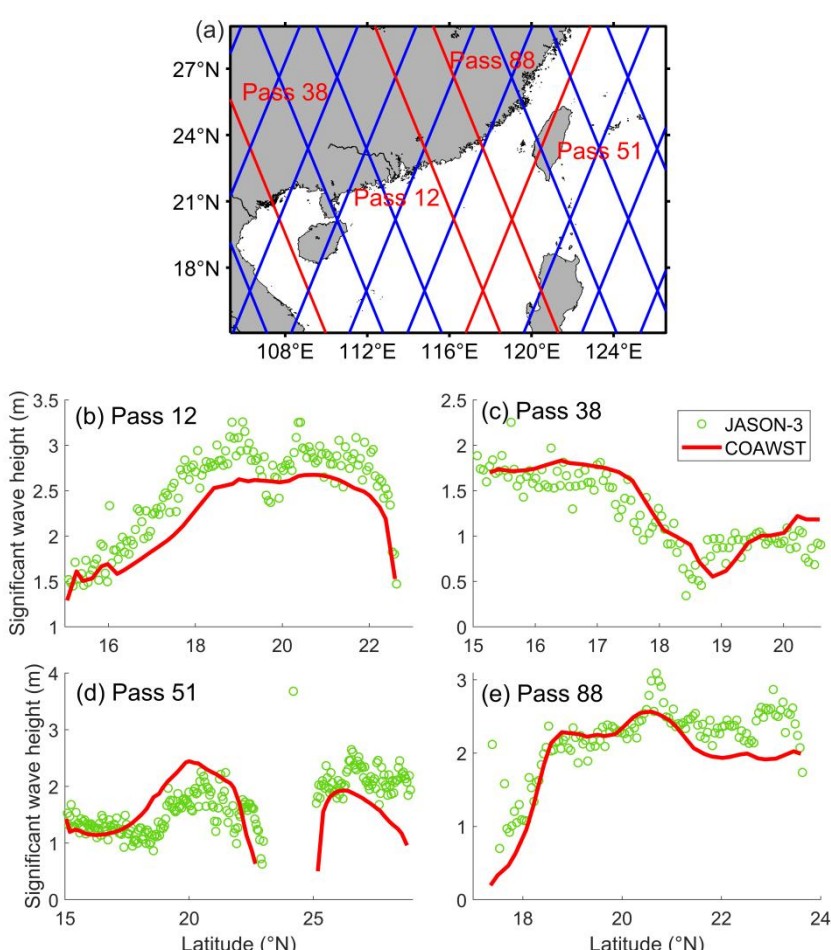

**Figure 9.** (a) Jason-3 ground track in the study area, where the red solid lines represent the tracks are in the period from 0000 UTC 1 January to 0000 UTC 7 January 2019. SWH comparison between model results and Jason-3 data at (b) 2200 UTC 03 January (c) 2200 UTC 04 January (d) 1100 UTC 05 January (e) 2100 UTC 06 January 2019.

## 5.4 Simulation results

In this section, the differences in power output, wind speed deficits, and turbulent kinetic energy (TKE) between the FWFP and Fitch schemes are analyzed in a realistic case using the fully coupled atmosphere-wave model. The last 6 hours (1200 UTC 01 January to 1800 UTC 01 January) of simulations are averaged for all results shown below.

### 5.4.1 Power output and wind speed deficits

For the majority of wind turbines (93.2 % of the number of grids), the power output of WRF-Fitch is smaller than that of WRF-FWFP. This is most likely due to the fact that the FWFP scheme takes into account that the frictional velocity is

lower (the inflow wind speed is higher). The difference in power output of a grid cell can reach a maximum of 18 MW (Figure 10a), which means that the difference in power output of a single turbine can reach a maximum of 2 MW (about 9 turbines in a grid cell). In a small area (6.8 % of the number of grids, Figure 10c), the power output of WRF-Fitch is greater than that of WRF-FWFP, but the difference in power output can only reach a maximum of 0.06 MW (3 %). And most of the positive differences are located in the upstream grid cells (Figure 10b). On average, the power prediction in the FWFP scheme increases by 12 % compared to the Fitch scheme (Figure 10d).

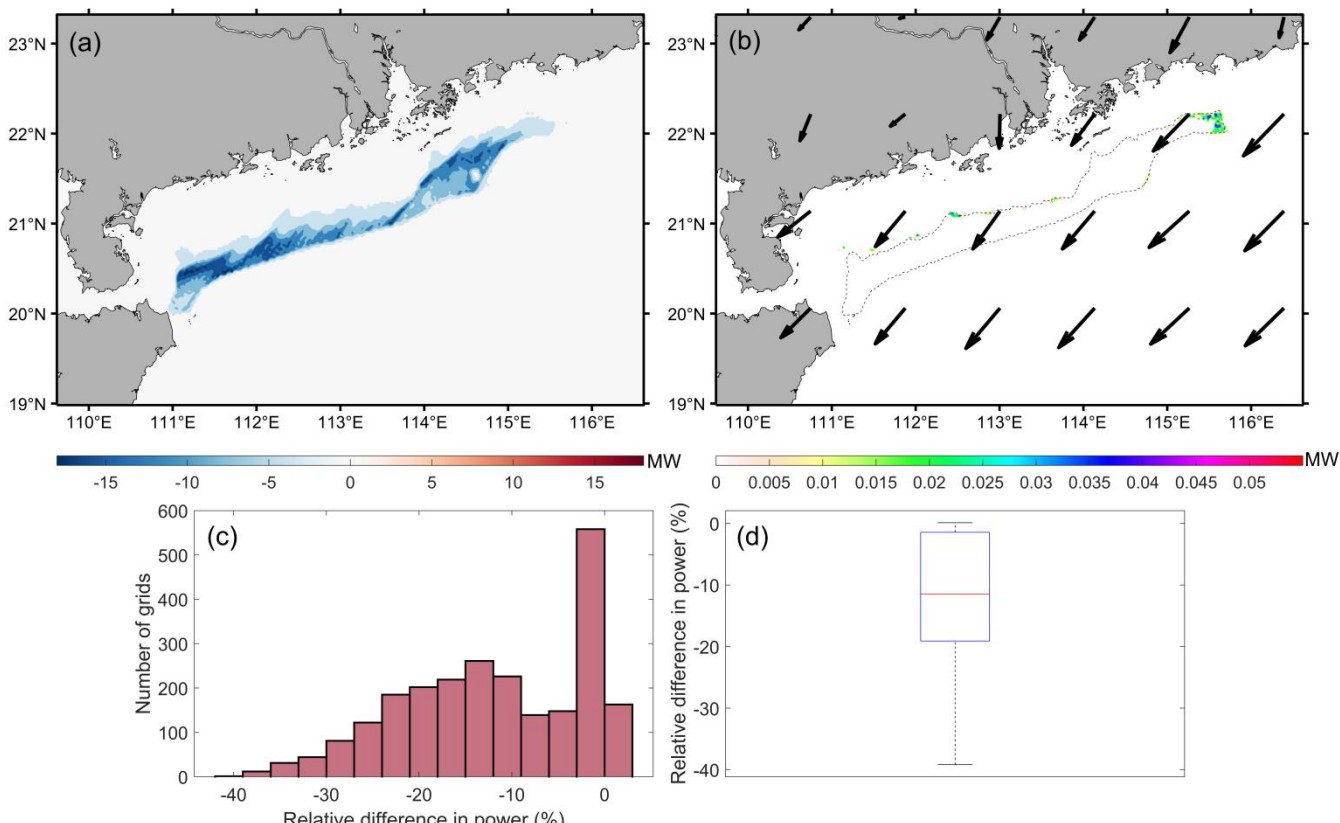

**Figure 10**. (a) (b) Power output of the WRF-Fitch case minus the WRF-FWFP case, but only positive values are shown in Figure 10b. The black dashed line indicates the outer boundary of the wind farm, and the black arrow indicates the wind direction. (c) Histogram of the relative difference between the power output of the WRF-Fitch case and the power output of the WRF-FWFP case. (d) Boxplot of the relative difference in the power output, the same as Figure 6.

The maximum value of momentum reduction at hub height for the FWFP scheme is 8 m/s (Figure 12a). Downstream of the wind farm, the wind speed deficit extends into a long wake. The length of the wake reaches 70 km for a wind speed deficit of 1 m/s (Figure 13a). Previous studies found that roughness lengths and turbulence intensity are lower when the subsurface of the atmosphere is oceanic. Therefore, the wake behind offshore wind farms is expected to be much longer than onshore (~50 km) (Emeis et al., 2016; Lundquist et al., 2019). The difference in wind speed at hub height between the two schemes has both positive and negative values. The negative values are mainly distributed on the upstream side of the wind

farm and in the middle of the wind farm, with a minimum value of about -0.4 m/s. The positive values areas show a more pronounced difference, up to a maximum of 1.8 m/s. These effects are also propagated to the wind farm wake (Figure 12b).

The largest uncertainty comes from the surface parameterization scheme in the WRF model. Semi-submersible floating wind turbines weaken the significant wave height, which leads to changes in frictional velocity and roughness length at the wind turbine site. In addition, the atmospheric stability $\varphi(\frac{z}{L})$ at this location also changes according to the Monin-Obukhov similarity theory (Figure 11). These factors together lead to a change in the inflow wind speed.

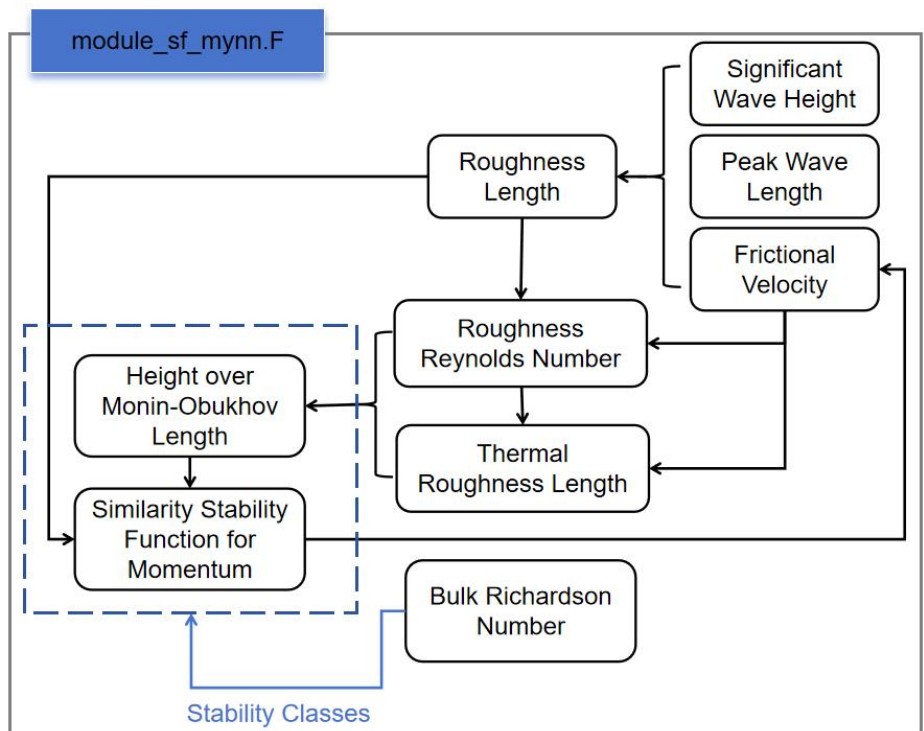

**Figure 11**. Flowchart of the computation of surface layer variables in the WRF model.

Vertical profiles of wind speed deficits in WRF-FWFP case also show similar characteristics to WRF-Fitch case. The atmospheric boundary layer (ABL, including downstream) is affected by the wind speed deficit caused by wind farms (Figure 13a). A wind speed deficit of 1 m/s can extend up to the top of the ABL. Figure 13b shows that the reduction in momentum of the FWFP scheme also spread to throughout the ABL, which is most pronounced in the rotor area with a

325 maximum value of 1.5 m/s. The top of the turbine to the top of the ABL and the wind-farm wakes also have an effect with values of 0.1 to 0.8 m/s. The maximum value of the vertical gradient of the wind speed difference is between the hub height and the top of the turbine (163 m).

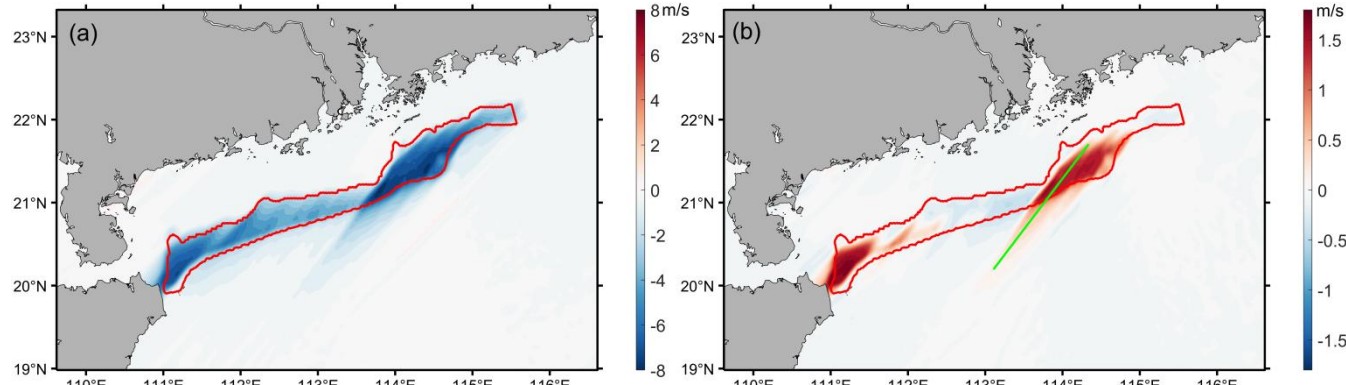

**Figure 12.** Horizontal wind speed of (a) the WRF-FWFP case minus the WRF-CTL case and (b) the WRF-Fitch case minus the WRF-FWFP case at the hub height level. The red solid line indicates the outer boundary of the wind farm, and the green solid line indicates a cross section analyzed further.

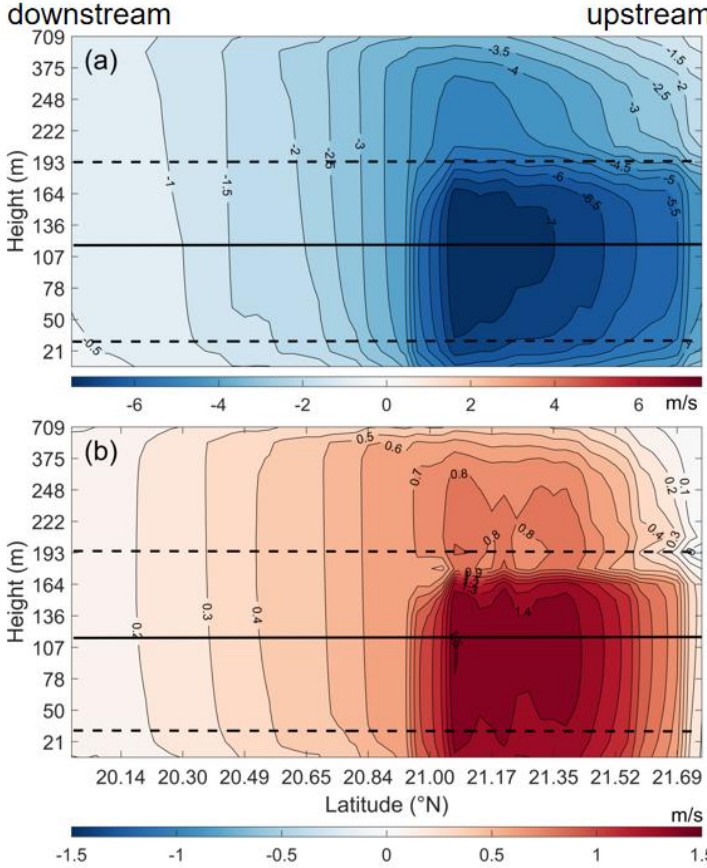

**Figure 13.** Vertical transect of the wind speed of (a) the WRF-FWFP case minus the WRF-CTL case and (b) the WRF-Fitch case minus the WRF-FWFP case along the green solid line in Figure 12b. The turbine hub height is indicated by the horizontal solid line and the turbine blade bottom and top by the dashed lines.

## 5.4.2 TKE

Despite the advection of TKE, it decays rapidly downstream. TKE generated within the wind farm is largely localized within the wind farm area. The maximum increase in TKE at the top of the turbines within the wind farm is 3.5 $m^2 s^{-2}$ (Figure 14a). The distribution of horizontal TKE differences is highly similar to that of horizontal wind speed differences (Figure 12b), indicating that the wind shear may dominate the TKE distribution. The reduction in the TKE caused by the wind farm continues to extend more than 100 km downstream near the surface (Figure 14c). In contrast, there are two localized areas within the wind farm where the TKE rises obviously, corresponding to the two areas with the greatest increase in TKE at the top of the turbine (Figure 14c). There is little difference in TKE between the WRF-Fitch and the WRF-FWFP in the near the surface downstream (Figure 14d), with most of differences occurring only within the wind farm. The distribution of differences in TKE near the ground is almost identical to that at the top of the turbine, except that the value is smaller, by about 0.9 $m^2 s^{-2}$ (60%), which is most likely due to the vertical transport of TKE. The reduction in TKE near the surface in the downstream (Figure 14c) is due to a wind speed deficit and corresponding decrease in wind shear in the lower levels of the wake, resulting in a decrease in shear production in TKE and the reduction in the TKE is no higher than at the top of the turbine (Fitch et al., 2012).

As the wind speeds decrease, the increase in TKE extends to the top of the ABL which is above the wind farm, with an increase of 1 $m^2 s^{-2}$ reaching a height of nearly 709 m (Figure 15a). At the top of the turbine, the maximum increase in TKE is 3.6 $m^2 s^{-2}$. On the other hand, the TKE below the bottom of the turbine blades decreases more and more as it approaches the sea surface. Figure 14b indicates that the largest effects of the FWFP scheme on the TKE also appear at the top of the turbine (-1.4~0.6 $m^2 s^{-2}$). It is evident that these results are not influenced by advection, but by vertical transport to spread throughout the ABL.

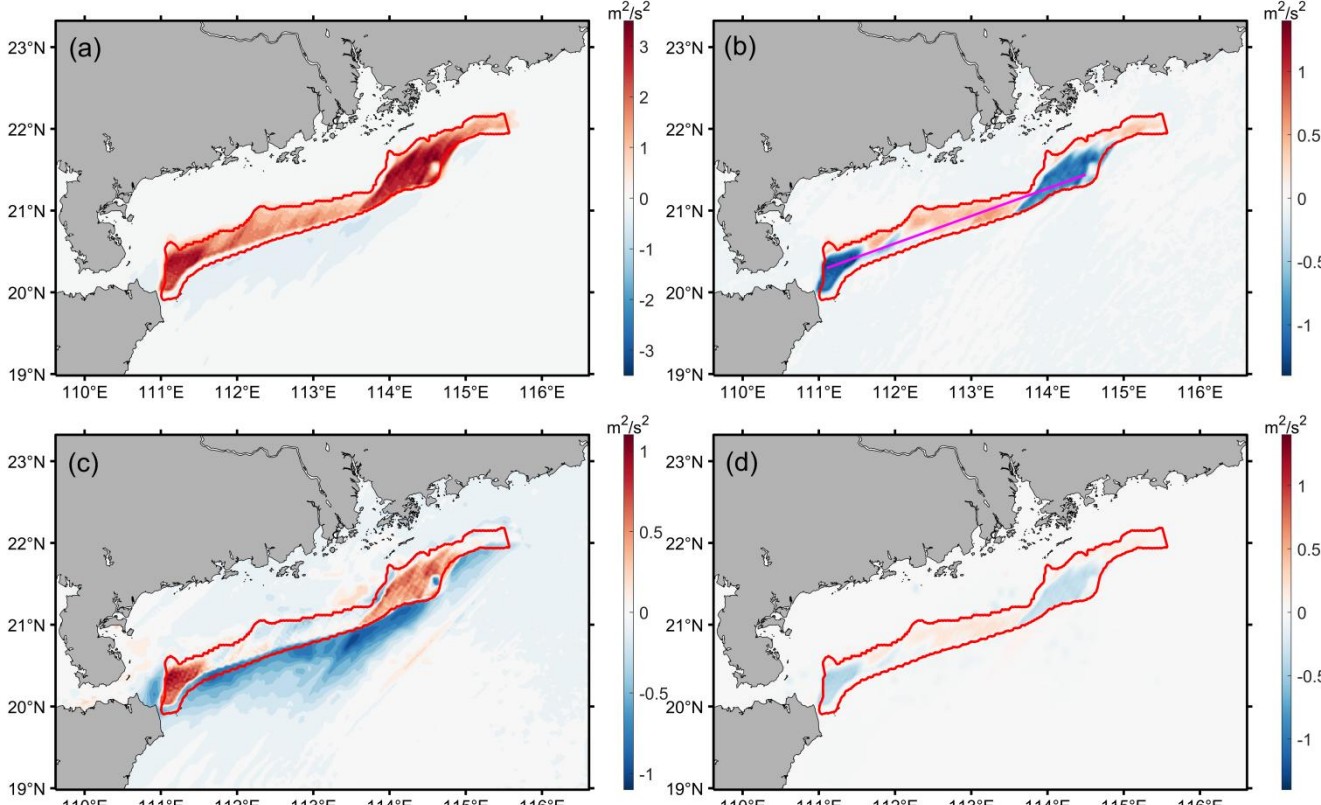

**Figure 14.** Horizontal TKE of (a) the WRF-FWFP case minus the WRF-CTL case and (b) the WRF-Fitch case minus the WRF-FWFP case at the top of the turbine. Horizontal TKE of (c) the WRF-FWFP case minus the WRF-CTL case and (d) the WRF-Fitch case minus the WRF-FWFP case near the sea surface. The red solid line shows the outer boundary of the wind farm, and the pink solid line indicates a cross section analyzed further.

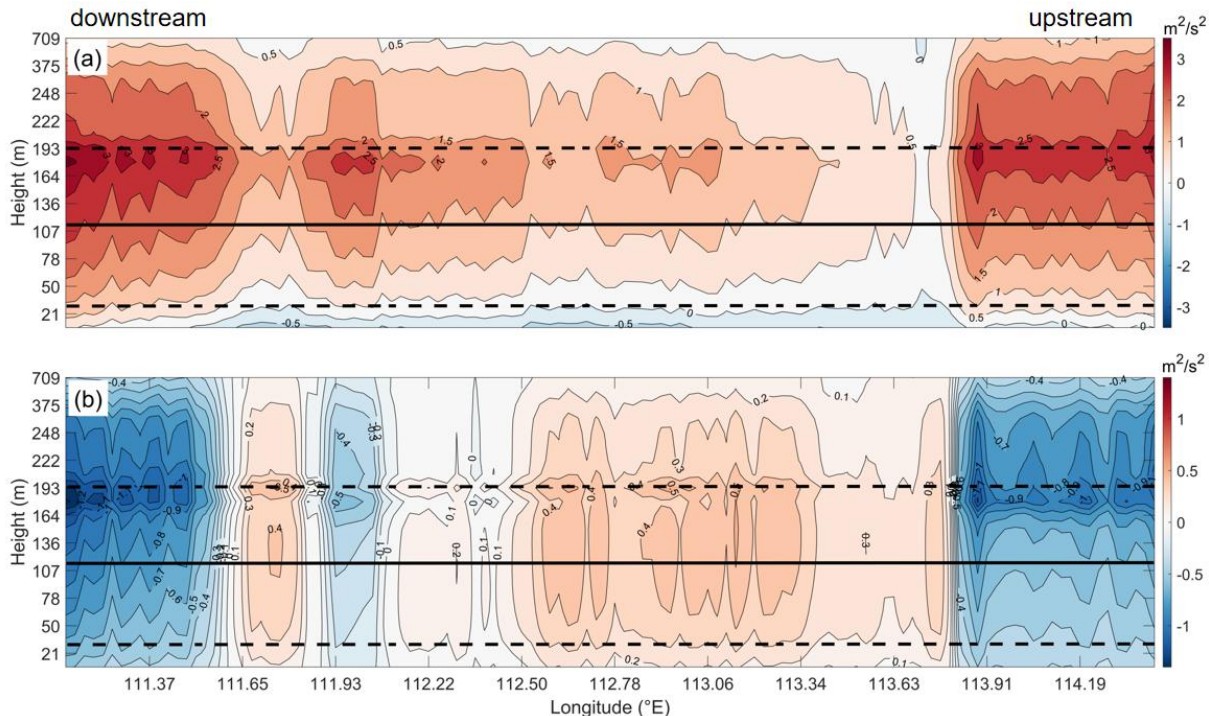

**Figure 15.** Vertical transect of the TKE of (a) the WRF-FWFP case minus the WRF-CTL case and (b) the WRF-Fitch case minus the WRF-FWFP case along the pink solid line in Figure 14b. The turbine hub height is indicated by the horizontal solid line and the turbine blade bottom and top by the dashed lines.

The changes in TKE generation due to the new wind farm parameterization are mainly due to variation in wind shear and vertical transport, which are further analyzed and quantified using the TKE budget. In the Planetary Boundary Layer scheme (MYNN 2.5), the TKE is expressed as:

$$\frac{\partial TKE}{\partial t} = P_s + P_b + P_v + P_d, \tag{28}$$

where $P_s$ is the shear production term, $P_b$ is the buoyancy production term, $P_v$ is the vertical transport term, and $P_d$ is the dissipation term. Details about the equations can refer to Janjic (2001).

The largest sources of the difference in TKE between the two cases are shear generation and vertical transport, with the dissipation term almost negligible (Figure 16). Vertical transport clearly affects the TKE from the bottom of the turbine blade to the sea surface and from the hub height to a height of 248 m. The TKE from above the turbine increases the TKE from the hub height to the top of the turbine due to downward vertical transport (Figure 16a). The intensification in TKE due to shear generation appears at the top of the turbine, while the reduction appears below the top of the turbine (150 to 194 m) (Figure 16b). In addition, the impacts of shear generation and vertical transport on the TKE in the rotor region are almost exactly opposite, but the effect of shear generation is slightly larger. This also corresponds well with the distribution of the differences in TKE (Figure 15b).

Subsequently, the positive and negative areas of the TKE on the cross section are analyzed separately. In the region
where the TKE decreases (the wind speed increases), the FWFP scheme results in a greater wind speed deficit, which
increases the vertical wind shear in the region from the hub height to the top of the wind turbine, thus increasing the TKE
(Figure 17a). In the region where the difference in TKE is positive (and the difference in wind speed is negative), the result
is the opposite (Figure 17b). And in these regions, the FWFP scheme recovers only a small increase in wind speed, which
also results in a limited reduction in TKE.

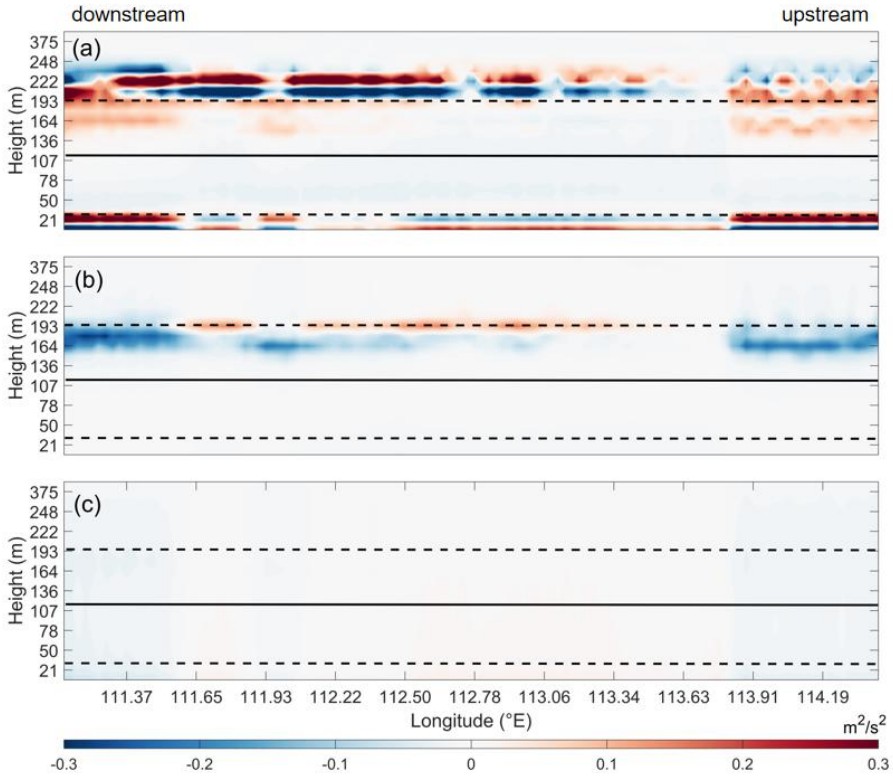

**Figure 16.** Vertical transect of the TKE budget components of the WRF-Fitch case minus the WRF-FWFP case: (a) vertical transport (b) shear generation (c) dissipation along the pink solid line in Figure 14b. The turbine hub height is indicated by the horizontal solid line and the turbine blade bottom and top by the dashed lines.

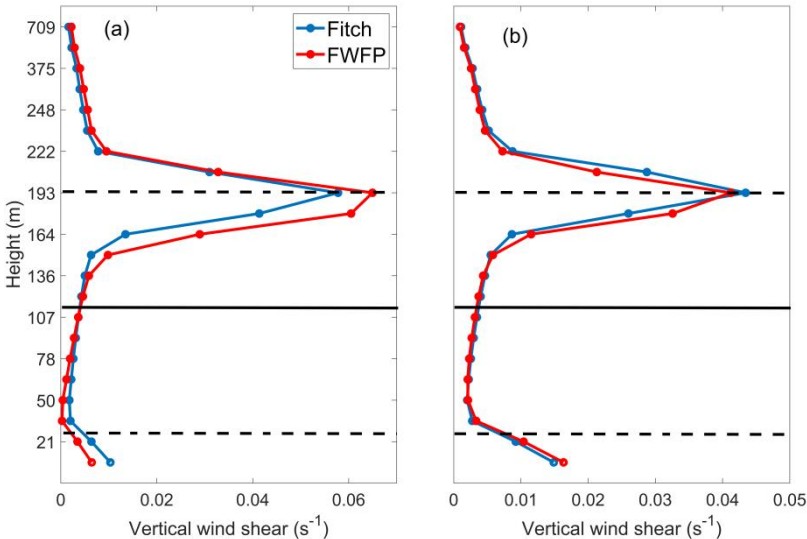

**Figure 17.** Profile of the vertical wind shear along the pink solid line (within the wind farm) in Figure 14b (a) areas where the difference (the WRF-Fitch case minus the WRF-FWFP case) in TKE is negative (b) areas where the difference (the WRF-Fitch case minus the WRF-FWFP case) in TKE is positive. The turbine hub height is indicated by the horizontal solid line and the turbine blade bottom and top by the dashed lines.

## 6 Conclusions

Floating wind turbines are essential as the offshore wind industry moves into deeper ocean regions. However, current wind farm parameterizations can only be applied to fixed turbines. In this study, we develop a floating wind farm parameterization (FWFP) in a coupled model.

Parameters of the column is modified in the VEG module of SWAN to include the effect of the inertial force to make it suitable for the application of the side column of a floating offshore wind turbine. At the same time, a series of idealized
high-resolution SWAN simulations are conducted to investigate the dissipation of wave energy induced by the side columns of floating turbines. It is found that under certain conditions, the side columns of floating turbines can attenuate more than 50 % of the significant wave height (SWH), and a wave "wake" phenomenon occurs with a recovery length of ~1 km. The mean wave direction is also affected, with a symmetric change of about 5° around the side columns, and the mean wave length increases by more than 20 m. The idealized SWAN simulations and theoretical analyses show that the attenuation of
the SWH decreases with increasing water depth and is enhanced with increasing peak wave period. A total of 2520 groups of experiments consisting of different incident SWHs, water depths, and peak wave periods are conducted, and the results of these idealized simulations are used to train a Gaussian process regression (GPR) model with the Matern 5/2 kernel. This model can predict the attenuated SWH due to the side columns of the floating turbine with a given water depth, peak wave period, and incident SWH.

The GPR model is implemented in the surface parameterization module of the WRF to calculate the frictional velocity, roughness length, and other relevant variables at the wind turbine site. These variables are then passed to the Planetary Boundary Layer Driver module to calculate the new inflow wind speeds for the Fitch wind farm parameterization to form the FWFP. The difference in the results between the original Fitch scheme and our new FWFP scheme is analyzed in a realistic simulation using a coupled atmosphere-wave model. The results indicate that the FWFP scheme results in higher power output for most of the grid cells in the wind farm (93.2%) compared to the Fitch scheme, with an average of 12% higher. This is most likely due to the fact that floating wind farms weaken the SWH and also reduce the roughness length and frictional velocity, which ultimately leads to an increase in the inflow wind speed. There is also a significant difference in the wind speed deficit caused by the two schemes. As a result of the FWFP modification of the inflow wind speed, the wind speed in the upstream region increases (<0.4 m/s) compared to the Fitch scheme. Wind speeds in the downstream region are reduced, but to a greater extent, to within 1.8 m/s. The distribution of the differences in TKE corresponds well to the distribution of the differences in wind speed. Compared to the Fitch scheme, the FWFP scheme generates less TKE in the upstream region ($< 0.6$ m$^2$/s$^{-2}$) and more TKE in the downstream region ($< 1.4$ m$^2$/s$^{-2}$). The TKE budget demonstrates that shear generation dominates the difference in TKE between the two schemes in the rotor region. Vertical transport is also an important source of TKE variation, and FWFP reduces TKE transport from above the top of the turbine to the lower levels.

*Code and Data availability statement.* The NCEP FNL data (NECP, 2015) can be download at website https://rda.ucar.edu/datasets/ds083.3/, wave data (WW3) can be download at https://www.ncei.noaa.gov/thredds-ocean/catalog/ncep/nww3/catalog.html (WW3DG, 2019), and the satellite data (Jason-3) can be obtained from https://www.ncei.noaa.gov/products/jason-satellite-products (Lillibridge, 2019). The coupled model is freely available online (https://github.com/DOI-USGS/COAWST) (Warner et al., 2010). Fortran files related to the offshore wind farms parameterization are available in the public repository (https://osf.io/arj3m/).

*Author contributions.* SD: Methodology, Formal analysis, Investigation, Writing - Original Draft, Writing - Review & Editing; SY: Formal analysis, Writing - Review & Editing; SC: Conceptualization, Funding acquisition, Formal analysis, Investigation, Writing - Review & Editing; DC: Funding acquisition, Writing - Review & Editing; XY: Data Curation, Investigation; SC: Investigation, Writing - Review & Editing

*Competing interests.* The authors declare that they have no known competing financial interests or personal relationships that could have appeared to influence the work reported in this paper.

*Acknowledgements.* This study is supported by funds from Shenzhen Science and Technology Innovation Committee (WDZC20200819105831001) and the Guangdong Basic and Applied Basic Research Foundation (2022B1515130006).

## Appendix

### a. Linear regression

Linear regression models are the simplest and most basic class of supervised learning models in machine learning. Stepwise linear regression is an effective method when there are more independent variables. The idea of stepwise linear regression is to introduce the independent variables one by one, and for each independent variable introduced, the F-test is performed one by one, and when the originally introduced variable becomes insignificant due to the introduction of the later introduced variable, it is dropped, and the process is repeated until the regression equation contains only significant regression variables.

(1) Create a one-way regression equation for each independent variable with the dependent variable.

$$y = a_i X_i + b_i , i = 1,2,\ldots,m \tag{a1}$$

(2) Calculate the test statistic F for the regression coefficients in each regression equation separately and find the maximum value $F_{k_1}^1 = max\{F_1^1, F_2^1, \ldots, F_m^1\}$. If $F_{k_1}^1 \leq F_\alpha(1, n-2)$, stop filtering, otherwise select $x_{k_1}$ in the set of variables, at this point consider $x_{k_1}$ as $x_1$ and proceed to step (3).

(3) Separately, the set of independent variables $(x_1, x_2)$, $(x_1, x_3)$,..., $(x_1, x_m)$ with the dependent variable to create a binary regression equation. Return to step (2) and process the $m - 1$ regression equations. This iterative process results in the optimal regression equation.

### b. Regression tree

Decision trees are a non-parametric supervised learning method for classification and regression. The goal is to create a model that predicts the value of a target variable by learning simple decision rules derived from data features. Depending on the type of data being processed, decision trees fall into two categories: classification decision trees, which can be used to process discrete data, and regression decision trees, which can be used to process continuous data. In the input space where the training data set is located, binary decision trees are constructed by recursively dividing each region into two sub-regions and determining the output values on each sub-region. Suppose that the input space has been divided into $M$ cells $(R_1, R_2, .., R_M)$ and that each cell $R_m$ has a fixed output value $c_m$.

(1) Select the optimal cutoff variable $j$ and cutoff point $s$ so that Eq. (b1) is minimized.

$$min[min \sum_{x_i \in R_1(j,s)} (y_i - c_1)^2 + min \sum_{x_i \in R_2(j,s)} (y_i - c_2)^2] \tag{b1}$$

(2) Divide the region with the selected pair $(j, s)$ and decide the output value of the response.

$$R_1(i, s) = \{x|x^j \leq s\}, R_2(i, s) = \{x|x^j > s\} \tag{b2}$$

$$\hat{c}_m = \frac{1}{N_m} \sum_{x_i \in R_m(j,s)} y_i, x \in R_m, m = 1, 2 \tag{b3}$$

(3) Repeat steps (1), (2) for both sub-regions until the condition is satisfied.

(4) The input space is divided into $M$ regions $(R_1, R_2, .., R_M)$ to generate a decision tree.

$$f(x) = \sum_{m-1}^{M} \hat{c}_m I \ (x \in R_m) \tag{b4}$$

### c. Support vector machines

Support vector regression (SVR) is a regression algorithm based on the support vector machine (SVM) for solving regression problems. Unlike traditional regression algorithms, the goal of SVR is to minimize the difference between predicted and actual values by constructing a prediction function. In addition, unlike general regression, SVR allows the

model to have a certain amount of deviation, and the points within the deviation range are not considered problematic by the model, while the points outside the deviation range are counted as losses.

For SVR, there is a prediction model as follows

$$f(x) = w^T x + b \tag{c1}$$

The optimization objective function is as follows

$$min \, J = \frac{1}{2} \|w\|^2 \tag{c2}$$

subject to $|y_i - w^T x_i - b| \leq \varepsilon, \, i=1,2,...,N$

To eliminate the influence of possible singular value data on the performance of the SVR model, a slack variable $\delta$ is introduced and Eq. (c2) is transformed into

$$min \, J(w, \delta, \delta_i^*) = \frac{1}{2} \|w\|^2 + c \sum_{i=1}^{N} (\delta_i + \delta_i^*) \tag{c3}$$

subject to $y_i - w^T x_i - b \leq \varepsilon + \delta_i^*$,

$w^T x_i + b - y_i \leq \varepsilon + \delta_i$,

$\delta_i, \delta_i^* \geq 0$

Among them, $c$ is the penalty parameter, the larger the value of $c$ indicates that the SVR regression model is less adaptive; a small value of $c$ will lead to a decrease in the sensitivity of $\delta$, increasing the training error. The penalty parameter

$c$ is a balanced compromise between the complexity and the adaptability of the SVR model and has to be set in the application. It can be seen that the original objective function is a quadratic programming problem. To facilitate the solution, the Lagrange function is introduced.

$$min \, L_p = \frac{1}{2} \|w\|^2 + c \sum_{i=1}^{N} \alpha_i (\varepsilon + \delta_i - y_i + w^T x_i + b) - \sum_{i=1}^{N} \alpha_i^* (\varepsilon + \delta_i^* + y_i - w^T x_i - b) - \sum_{i=1}^{N} (\beta_i \delta_i + \beta_i^* \delta_i^*) \tag{c4}$$

where $\alpha_i, \alpha_i^*, \beta_i, \beta_i^* \geq 0$ are Lagrange multipliers. The optimization problem (c4) for the original objective function is

transformed into a saddle-point problem for solving the Lagrange function.

$$max \, L_D = -\frac{1}{2} \sum_{i=1}^{N} \sum_{j=1}^{N} (\alpha_i - \alpha_i^*)(\alpha_j - \alpha_j^*)\langle \varphi(x_i), \varphi(x_j) \rangle - \varepsilon \sum_{i=1}^{N} (\alpha_i + \alpha_i^*) + \sum_{i=1}^{N} y_i (\alpha_i - \alpha_i^*) \tag{c5}$$

subject to $\sum_{i=1}^{N} (\alpha_i - \alpha_i^*) = 0, \, \alpha_i \geq 0, \, \alpha_i^* \leq c$

The corresponding regression estimation function is obtained.

$$f(x) = \sum_{i=1}^{N} (\alpha_i - \alpha_i^*)\langle \varphi(x_i), \varphi(x_j) \rangle + b \tag{c6}$$

In practice, it is often difficult to define feature mappings for which kernel functions $K(x_i, x_j)$ are introduced.

$$f(x) = \sum_{i=1}^{N} (\alpha_i - \alpha_i^*)K(x_i, x_j) + b \tag{c7}$$

### d. Gaussian process regression

Gaussian Process Regression (GPR) is a nonparametric model for regression analysis of data using a Gaussian Process prior. GPR is usually used for low-dimensional and small-sample regression problems. In regression prediction, it usually predicts the value of a single point, but GPR can be interpreted as probabilistic prediction, which can predict the exact point as well as the upper and lower bounds, adding more reference value to the prediction. The model assumptions of GPR consist of two parts: the noise (regression residuals) and the Gaussian Process prior, and its solution is based on Bayesian inference. The GPR can be expressed as,

$$f(x) \sim N(\mu(x), k(x_i, x_j)) \tag{d1}$$

where $\mu(x)$ denotes the mean function, $k(x_i, x_j)$ denotes the kernel (covariance) function, showing that determining the mean and covariance functions is sufficient to determine a GPR. Kernel functions are at the core of GPR, and there are many different kernel functions. For example, the Matern 5/2 covariance function is defined as,

$$k(x_i, x_j) = \sigma_f^2 (1 + \frac{\sqrt{5}r}{\sigma_l} + \frac{5r^2}{3\sigma_l^2})exp(-\frac{\sqrt{5}r}{\sigma_l}) \tag{d2}$$

where $r = \sqrt{(x_i - x_j)^T((x_i - x_j))}$. The kernel parameters are based on the signal standard deviation $\sigma_f$ and the characteristic length scale $\sigma_l$.

The given discrete data is $(x^o, y^o)$. Suppose $y^o$ and $f(x)$ have a joint Gaussian distribution. Then the joint probability density formula is:

$$\begin{matrix} f(x) \\ y^o \end{matrix} \sim N(\begin{matrix} \mu_f \\ \mu_{y'} \end{matrix} \begin{matrix} K_{ff} & K_{fy} \\ K_{fy}^T & K_{yy} \end{matrix}) \tag{d3}$$

where $K_{ff} = k(x, x)$, $K_{fy} = k(x, x^o)$, $K_{yy} = k(x^o, x^o)$

The mean of the prediction is then.

$$y_{mean} = K_{fy}^T K_{yy}^{-1} y^o \tag{d4}$$

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
