# Peer review of "A parameterization scheme for the floating wind farm in a coupled atmosphere-wave model (COAWST v3.7)"

_EGUsphere, 2023_

## Referee Comment (RC2)

**Review: A parameterization scheme for the floating wind farm in a coupled atmosphere-wave model (COAWST v3.7)**

**Summary**

The authors implement a floating wind turbine parameterization in a coupled atmosphere-wave model. Their parameterization accounts for changes in wave properties due to the turbine's floating structure. In their wave parameterization, the authors develop a regression model, trained using a spectral wave model (SWAN), that accounts for the turbine's floating structure. The authors also modify the momentum tendency in the surface layer of the atmosphere. A source of momentum is included in the momentum tendency equation to represent changes in the momentum flux due to the floating turbine. Finally, the authors compare their floating turbine parameterization against the Fitch parameterization for a wind farm in the South China Sea.

The manuscript addresses a very interesting topic, namely the importance of including coupled atmosphere-wave models to evaluate the effects from offshore wind turbines in the flow over large regions. However, I have major concerns that should be addressed prior to publication, mainly about their modifications to the Fitch wind farm parameterization, which adds a non-physical source of momentum across the surface layer.

**Major comments:**

1. Machine learning models: the manuscript lacks information about the ML models used therein. Also, there is no explanation of how the data are split into training and validation. Specifically:
    a. The authors mention four different machine learning models. However, they do not provide information about neither of these models. Please include a more thorough description of each model, perhaps as an Appendix.
    b. It seems the authors are training and validating the models using the same dataset. If so, this should be revised; otherwise, it is expected that the ML models are going to perform well. If not, please explain how you split the data for model validation.
2. Momentum source across the surface layer (section 5.1): the authors include a non-physical source of momentum at turbine heights. Specifically:
    a. I agree that changes in the momentum flux caused by variation in SWH affect winds close to the surface. However, these changes should be transmitted through modifications to the wall model (like in Jenkins et al., 2012; Paskyabi et al., 2014; Porchetta et al., 2021; Wu et al., 2020; Zou et al., 2018) rather than as an explicit source of momentum in the tendency equation over the bottom half of the turbine rotor layer.
    b. What is the reasoning behind adding non-physical sources of momentum to across the surface layer? Also, shouldn't the source of momentum decay with

height? If this is the case, then this should be rephrased as a modified wall model.

    c. The references provided in Lines 41-43 suggest waves modify the wind profile through changes in surface stresses, not through injections of momentum across the surface layer: AlSam et al. (2015) and Yang et al. (2014) study how swell can modify wake propagation. Jenkins et al. (2012) use a coupled atmosphere-ocean model that modifies the wind field through changes in surface roughness. Kalvig et al. (2014) resolve waves with a moving mesh, thus the wind profile is effectively modified by changes in surface roughness. Paskyabi et al. (2014) develop a wall model that accounts for wave-induced momentum fluxes. Porchetta et al. (2021) and Wu et al. (2020) use an atmosphere-wave coupled model, where the winds are modified by waves through changes in surface roughness. Zou et al. (2018) also focuses on a wall model.

3. Model configuration in Section 5.2: The authors use a 12 km horizontal grid spacing for their simulations. However, Tomaszewski and Lundquist (2020) show such coarse grids can produce unrealistic impacts over a very broad region. Please explain your choice of grid spacing.

4. Section 5.4: The authors conclude that Fitch overestimates wake effects. However, the FWFP is artificially accelerating wake recovery downstream of the turbines. Thus, it is expected to have higher power production estimates and lower wake deficits in the FWFP.

    a. Lines 243-244: Adding a source to the momentum tendency is expected to accelerate wake recovery downstream of the turbines. Thus, is it reasonable to say that that Fitch underestimates power output? Rather, the momentum source in the FWFP accelerates wake recovery; thus, momentum availability increases amplifying power production.

    b. Lines 257-258: same as above.

**Minor comments:**

1. I recommend English language revisions throughout the manuscript.
2. Lines 22-24: What about coupled meso-microscale simulations? Coupled mesoscale-LES simulations using WRF can capture these effects, however, at a higher computational cost.
3. Line 31: Please add punctuation as: "… sink on the mean flow. Most of …"
4. Lines 27-44: I recommend splitting paragraph #2 in the introduction, perhaps at line 35.
5. Lines 43-44: I would argue that the current parameterization can be suitable for floating offshore wind farms. Rather, the atmosphere-only model in WRF does not capture changes in roughness length over the ocean caused by the presence of floating turbines.
6. Line 74: Please explain why you chose d = 20 m.
7. Captions should fully describe the figure. Please include additional information in all captions to make each figure self-explanatory. For example, include description of the different terms and symbols used in Figure 1, as well as the significance of the red contours.

8. Figure 3: It is difficult to read the information within the grey area. Please use colors with higher contrast. Also, what is the meaning of the blue curves (presumably schematic for waves) to the side of the plot?
9. Line 173: "The important point in the derivation …" implies that the source of TKE in the Fitch parameterization is not important. Please rephrase.
10. Lines 193-201 and Figure 6: Please maintain consistency in your nomenclature (e.g., the authors use $u_{*,wt}$ in Eq. 17, but ustwt in Figure 6)
11. Figures 11, 13, 14: It would be helpful to show the top and bottom of the turbine rotor layer for reference.

**References:**

AlSam, A., Szasz, R., and Revstedt, J.: The Influence of Sea Waves on Offshore Wind Turbine Aerodynamics, Journal of Energy Resources Technology, 137, 051209, https://doi.org/10.1115/1.4031005, 2015.

Jenkins, A. D., Paskyabi, M. B., Fer, I., Gupta, A., and Adakudlu, M.: Modelling the Effect of Ocean Waves on the Atmospheric and Ocean Boundary Layers, Energy Procedia, 24, 166–175, https://doi.org/10.1016/j.egypro.2012.06.098, 2012.

Kalvig, S., Manger, E., Hjertager, B. H., and Jakobsen, J. B.: Wave Influenced Wind and the Effect on Offshore Wind Turbine Performance, Energy Procedia, 53, 202–213, https://doi.org/10.1016/j.egypro.2014.07.229, 2014.

Paskyabi, M. B., Zieger, S., Jenkins, A. D., Babanin, A. V., and Chalikov, D.: Sea Surface Gravity Wave-wind Interaction in the Marine Atmospheric Boundary Layer, Energy Procedia, 53, 184–192, https://doi.org/10.1016/j.egypro.2014.07.227, 2014.

Porchetta, S., Muñoz-Esparza, D., Munters, W., Van Beeck, J., and Van Lipzig, N.: Impact of ocean waves on offshore wind farm power production, Renewable Energy, 180, 1179–1193, https://doi.org/10.1016/j.renene.2021.08.111, 2021.

Tomaszewski, J. M. and Lundquist, J. K.: Simulated wind farm wake sensitivity to configuration choices in the Weather Research and Forecasting model version 3.8.1, Geosci. Model Dev., 13, 2645–2662, https://doi.org/10.5194/gmd-13-2645-2020, 2020.

Wu, L., Shao, M., and Sahlée, E.: Impact of Air–Wave–Sea Coupling on the Simulation of Offshore Wind and Wave Energy Potentials, Atmosphere, 11, 327, https://doi.org/10.3390/atmos11040327, 2020.

Yang, D., Meneveau, C., and Shen, L.: Effect of downwind swells on offshore wind energy harvesting – A large-eddy simulation study, Renewable Energy, 70, 11–23, https://doi.org/10.1016/j.renene.2014.03.069, 2014.

Zou, Z., Zhao, D., Zhang, J. A., Li, S., Cheng, Y., Lv, H., and Ma, X.: The Influence of Swell on the Atmospheric Boundary Layer under Nonneutral Conditions, Journal of Physical Oceanography, 48, 925–936, https://doi.org/10.1175/JPO-D-17-0195.1, 2018.

---

## Author Comment (AC2)

*The submitted manuscript describes a new parameterization of the impact of wind farms on the wind speed and TKE within the boundary layer, whose novelty is that it is adapted to the characteristics of floating offshore turbines. As wind farms move to deeper waters, necessitating this design, such a parameterization will certainly be of great scientific value.*

*There is a lot of innovative analysis here, such as the consideration of impacts on wave heights, inertial effects from the piles, and the adaptation of a SWAN vegetation model to account for these effects. The machine learning approach is also interesting. However, overall I find the manuscript leaves out too many details of the theoretical justification, is unclear in presentation and organization (especially of the experimental design), and seems to provide insufficient evidence to justify its conclusions. So I can only recommend acceptance after major revisions have been performed.*

**Responses to the comments of Reviewer #1:**

**We sincerely thank the reviewer for the suggestions and comments that help us improve the quality of our manuscripts.**

*Comment 1: Line 27: You mention explicit and implicit methods here, and then state that explicit methods are superior, but you never describe what an implicit method is – you should put in either a brief description or remove the first sentence of the paragraph.*

**Response:** Thank you for your suggestion. We briefly describe the implicit method for parameterized wind farms in line 31 of the revised manuscript.

*Comment 2: Line 43: 'This suggests that the current wind farm parameterization is not suitable for floating wind farms because it does not account for the change in roughness length caused by large floating platforms.' Do you have additional evidence that existing parameterizations are deficient? Why would this deficiency only affect floating wind farms, and not also offshore but fixed wind farms?*

**Response:** Thank you for your suggestion. As described in lines 34 to 36 of the manuscript, almost all current studies on wind farm parameterization focus only on sub-grid effects of wind turbines. Additional evidence, in our view, can still come from the equation derived in Fitch et al. (2012),

$$\frac{\partial V_{ijk}}{\partial t} = \frac{-\frac{1}{2} N_{ij} C_T V_{ijk}^2 A_{ijk}}{z_{k+1} - z_k}$$

From the above equation, it can be seen that the momentum tendency term is only related to the number of turbines in the grid, the thrust coefficient, the inflow wind speed, the rotor area, and the vertical resolution. If the surface roughness of the turbine location is changed, the results obtained from the above equation do not change.

We also believe that the new wind farm parameterization proposed in this study can

also be applied to fixed offshore wind farms in a special case (large-diameter monopile foundation of offshore wind turbine). However, the diameter of the pile foundation of most of the current fixed wind turbines is only in the range of 1 to 4 meters (Figure R1). The weakening effect of such small diameter piles on the significant wave height is not obvious.

[Figure]

[Figure]

Figure R1 Wind turbine at Choshi demonstration site and its dimensions (Ishihara and Qian, 2018).

*Comment 3: Lines 80 and following: Maybe include more brief descriptions of where all these equations and values are coming from, and why?*

**Response:** Thank you for your suggestion. We apologize for missing some details. We have added relevant content in Section 2.

*Comment 4: Line 102: I don't follow the Rayleigh equations. Why would H^3 equal its integral over p(H) dH?*

**Response:** Thank you for your suggestion. Sorry for the confusion, we made changes in lines 123 to 130 of the revised manuscript. It is only when calculating the average energy dissipation that $H^3$ can be replaced by $\int_0^\infty H^3 p(H) dH$ in equation (11).

*Comment 5: Line 160: Maybe this works for the South China Sea, but a range of water depth from only 53 m to 98 m leaves out other potential deep water applications, such as off the U.S. West Coast, where the depth could be hundreds of meters (though at some point additional depth won't matter, I suppose).*

**Response:** This is a good comment, and we've taken it into account in our experimental design. In fact, using the high-resolution SWAN, we can set arbitrary water depths for ideal SWAN simulations. However, we limited the water depth to 100 meters for two reasons.

1) Considering the practical situation. Offshore floating wind turbines are still in their infancy, and it is challenging to achieve large-scale development of floating wind energy in areas with water depths of less than 100 meters.

2) The problem of excessive data volume. In this research, we use a data-driven

machine learning model and implement it in WRF. However, this also leads to the consumption of more computational resources. For other sea areas, we need to retrain our machine learning model. In this study, we simply chose a water depth in the range of 53 to 98 meters to save computational resources. How the effect of the floating wind turbine piles on the waves would vary with water depth (>100 meters) is also an interesting topic.

*Comment 6: Line 162: Your machine learning output variable is SWH, but roughness can also be a function of wavelength and wave age – would it be possible to include these parameters as well?*

**Response:** This is a good comment. In other roughness parameterization schemes, roughness is also correlated with wave age. When WRF is coupled to SWAN, the user can choose one of the following three parameterization schemes.

1. COARE_TAYLOR_YELLAND

$$z_0 = max(1200 * H * (\frac{H}{L + 0.001})^{4.5} + 0.11 * \frac{v}{u_* + 0.001}, 1.59 \times 10^{-5})$$

2. COARE_OOST

$$c = max(\frac{L}{P + 0.001}, 0.1)$$

$$z_0 = max(\frac{25}{\pi} * L * min(\frac{u_*}{c}, 0.1)^{4.5} + 0.11 * \frac{v}{u_* + 0.001}, 1.59 \times 10^{-5})$$

3. DRENNAN

$$c = max(\frac{L}{P + 0.001}, 0.1)$$

$$z_0 = max(3.35 * H * min(\frac{u_*}{c}, 0.1)^{3.4} + 0.11 * \frac{v}{u_* + 0.001}, 1.59 \times 10^{-5})$$

where $L$ is peak wave length, and $P$ is peak wave period. The effect of the turbine piles on the wave energy is only in the effective wave height and has no effect on $L$ and $P$. Therefore, the machine learning model only needs to output the significant wave height as a sole variable.

*Comment 7: Line 164: For non-specialists, could you include at least a little more description of what these ML methods mean? What is 'Matern 5/2 kernel'? Any explanation why its fit is so much closer than for the other methods?*

**Response:** Thank you for your suggestion. We apologize for missing some details. We describe each category of machine learning models in more detail in the Appendix.

The 'Maternal 5/2 kernel' is a kernel (covariance) function in Gaussian process regression (GPR).

Most GPR models have good performance, probably because the advantages of GPR is mainly in dealing with nonlinear and small sample data.

*Comment 8: Line 184: My biggest issue might be with the justification of (17). Why*

*does a change in surface momentum flux get translated to a change in mean wind kinetic energy for an elevated layer? Aren't changes in wind speed related to vertical gradients of momentum fluxes? And changes in kinetic energy related to the product of momentum flux times vertical gradients of mean winds? Why would the impact of surface roughness be omitted above 100 m if it can exert a drag on the whole boundary layer?*

**Response:** This is a good comment. We will first discuss with you about wind speed (kinetic energy) and momentum flux. According to the Monin-Obukhov similarity theory (Monin, 1954), assuming that the momentum flux in the near-surface layer of the atmosphere is constant, the vertical gradient of the wind speed in this layer can be expressed by the following equation,

$$\frac{kz}{u_*}\frac{du}{dz} = \varphi(\frac{z}{L})$$

$$\frac{kz}{u_*}\frac{du^2}{dz} = 2u \cdot \varphi(\frac{z}{L})$$

You are certainly right about the relationship between wind speed (kinetic energy) and momentum flux. But the reason we did not use the above equation in our study is that the algorithm for $\varphi$ in WRF is too complicated. We decided to use instead the equation derived by Fitch et al. (2012), which includes the unaccounted for wind stress (friction) that does the work.

And in this study, we believe that the reason the new parameterization only applies to heights below 100 m (Figure R2) is because this is approximately the maximum height of the near-surface layer (constant flux layer: momentum flux, heat flux, etc. vary less with height).

[Figure]

Figure R2 Composition of the atmospheric boundary layer.

*Comment 9: Line 230: You validate SWH with satellite data – are there any in situ measurements of waves available?*

**Response:** This is a good comment. We have had difficulty obtaining station observations. However, there have been numerous studies that have verified the accuracy of *Jason* satellite altimeter products.

*Comment 10: Line 231: You say the total model simulation time is 18 hours, but here you say the model is run for an additional 2 days for further validation, and then you show figures of model output over apparently four days. You also mention 'winter scenario' (line 243) but there is no mention of other scenarios. Can you clarify the simulation periods used in the evaluation, and state the relevant times in the figure captions?*

**Response:** Thank you for your suggestion. To validate the simulated SWH, we used *Jason-3* satellite data. However, if we had simulated for too short a period, it is likely that there would have been no satellite data available during that time to validate the SWH in the study area. Therefore, we simulated for a total of 6 days. This also helped to show that the coupled model performed well. We have made this clear in the revised manuscript. We also rewrote the caption of Figure 8. In addition, we have emphasized in line 287 that all later analyses are based on the simulations from 0600 UTC 1 January to 1200 UTC 1 January. We have redrawn Figure 8 to make it clearer.

*Comment 11: Line 250: In the caption 'Power output differences' – between what? I assume this is default Fitch – new FWFP scheme. But then it is incorrect to call these 'underestimates' of the power output because you don't know what the truth is, they are sensitivities.*

**Response:** Thank you for your suggestion. We have rewritten the caption of Figure 9, and what it shows is indeed the difference between the Fitch and FWFP schemes. The term 'underestimates' is indeed incorrect, and our purpose was indeed to examine the sensitivity of power output, wind speed deficit, and TKE to the FWFP scheme. We have therefore modified the wording in the latter analysis.

*Comment 12: Line 351: The first mention of Taylor and Yelland belongs in the experimental design, not at the end of the conclusions. I also would not agree that it is a 'complex iterative computational method'. It is a simple expression of roughness length as a function of SWH and wavelength (not frictional velocity), unless I am missing something?*

**Response:** This is a good comment. In other words, we consider the Taylor and Yelland scheme to be a 'complex iterative computational method' when using the coupled atmosphere-wave model. The Figure R3 shows the Taylor and Yelland expression from WRF/phys/module_sf_mynn.F. It can be seen that the roughness length is related to the significant wave height (HWAVE), the peak wave length (LWAVEP) and the frictional velocity (UST).

Figure R3 Taylor and Yelland expression in WRF.

The Figure R4 shows that the roughness length is in turn used to calculate the frictional velocity, which in turn is calculated to obtain the surface heat flux, moisture flux, and etc. The new frictional velocity is then passed through the loop code to the next roughness length calculation using the Taylor and Yelland expression. So we consider this a complex iterative algorithm to implement in a numerical model.

```
!-------------------------------------------------------------
!-----COMPUTE THE FRICTIONAL VELOCITY:
!-------------------------------------------------------------
!     ZA(1982) EQS(2.60),(2.61).
PSIX=GZ1OZ0(I)-PSIM(I)
PSIX10=GZ10OZ0(I)-PSIM10(I)
! TO PREVENT OSCILLATIONS AVERAGE WITH OLD VALUE
OLDUST = UST(I)
UST(I)=0.5*UST(I)+0.5*KARMAN*WSPD(I)/PSIX
!NON-AVERAGED: UST(I)=KARMAN*WSPD(I)/PSIX

! Compute u* without vconv for use in HFX calc when isftcflx > 0
WSPDI(I)=MAX(SQRT(U1D(I)*U1D(I)+V1D(I)*V1D(I)), wmin)
IF ( PRESENT(USTM) ) THEN
   USTM(I)=0.5*USTM(I)+0.5*KARMAN*WSPDI(I)/PSIX
ENDIF

IF ((XLAND(I)-1.5).LT.0.) THEN          !LAND
   UST(I)=MAX(UST(I),0.005)  !Further relaxing this limit - no need to go lower
   !Keep ustm = ust over land.
   IF ( PRESENT(USTM) ) USTM(I)=UST(I)
ENDIF
```

Figure R4 Code in WRF related to the calculation of frictional velocity.

*Technical corrections:*

*Note that this whole manuscript could greatly benefit by a technical edit for English usage.   I have only indicated the most noteworthy instances below.*

*Line 35:   'The installed capacity of offshore wind energy…'   This should be the beginning of a new paragraph.   The previous sentence does not clearly relate to the rest of the paragraph.*

**Response:** Thank you for your suggestion. We have split the second paragraph into two parts.

*Line 36:   change 'offshore' to 'near shore'.*

**Response:** Thank you for your suggestion. We have modified it, in line 40 of the revised manuscript.

*Line 71:   'Morrison' should be 'Morison'.*

**Response:** Thank you for your suggestion. We have modified it, in line 77 of the revised manuscript.

*Line 225:   In Table 1, 'Duhia' should be 'Dudhia', 'CORE' should be 'COARE', 'Talyor' should be 'Taylor'.*

**Response:** Thank you for your suggestion. We have modified the relevant contents of

Table 1.

We thank you again for giving us an opportunity to revise this manuscript, and look forward to hearing from you.

Sincerely,

Shengli Chen

---

## Author Comment (AC3)

*The authors implement a floating wind turbine parameterization in a coupled atmosphere-wave model. Their parameterization accounts for changes in wave properties due to the turbine's floating structure. In their wave parameterization, the authors develop a regression model, trained using a spectral wave model (SWAN), that accounts for the turbine's floating structure. The authors also modify the momentum tendency in the surface layer of the atmosphere. A source of momentum is included in the momentum tendency equation to represent changes in the momentum flux due to the floating turbine. Finally, the authors compare their floating turbine parameterization against the Fitch parameterization for a wind farm in the South China Sea.*

*The manuscript addresses a very interesting topic, namely the importance of including coupled atmosphere-wave models to evaluate the effects from offshore wind turbines in the flow overlarge regions. However, I have major concerns that should be addressed prior to publication, mainly about their modifications to the Fitch wind farm parameterization, which adds a non.physical source of momentum across the surface layer.*

**Responses to the comments of Reviewer #2:**

**We sincerely thank the reviewer for the suggestions and comments that help us improve the quality of our manuscripts.**

*Major comments:*

*Comment 1: Machine learning models: the manuscript lacks information about the ML models used therein. Also, there is no explanation of how the data are split into training and validation. Specifically:*

*a.  The authors mention four different machine learning models. However, they do not provide information about neither of these models. Please include a more thorough description of each model, perhaps as an Appendix.*

*b.  It seems the authors are training and validating the models using the same dataset. if so, this should be revised; otherwise, it is expected that the ML models are going to perform well. If not, please explain how you split the data for model validation*

**Response:** Thank you for your suggestion. We apologize for missing some details. We describe each category of machine learning models in more detail in the Appendix.

We realized previously that we were not defining some of the data as validation data. So we made a modification. The SWH is taken from 2 m to 4 m with 0.1 m interval. The peak wave period is from 7.4 s to 7.6 s, 8.4 s to 8.6 s, 9.6 s to 9.8 s, 11.0 s to 11.2 s with an interval of 0.1 s, and the water depth is selected from 53 m to 98 m with an interval of 5 m. This has a total number of 2520 (21 × 12 × 10) experimental groups. We then select simulated data that do not include water depths of 58 m, 78 m, 98 m to train several machine learning (regression) models, since data from these three water depths would be used as validation data. The result of the validation is shown below

(Figure R1), and the Matern 5/2 GPR model still performs best.

[Figure]

Figure R1. Box plots of RMSE for four typical ML regression models using validation data.The boxplots show the median (horizontal line), 25th to 75th percentile (box) and 5th to 95th percentile (whiskers). The whiskers extend to the most extreme data points not considered outliers, and the outliers are plotted individually using the ' + ' marker symbol.

*Comment 2: Momentum source across the surface layer (section 5.1): the authors include a non-physical source of momentum at turbine heights. Specifically:*

*a. I agree that changes in the momentum flux caused by variation in SWH affect winds close to the surface. However, these changes should be transmitted through modifications to the wall model (like in Jenkins et al, 2012; Paskyabi etal., 2014; Porchetta et al., 2021; Wu et al, 2020; Zou et al, 2018) rather than asan explicit source of momentum in the tendency equation over the bottom half of the turbine rotor layer.*

*b. What is the reasoning behind adding non-physical sources of momentum to across the surface layer? Also, shouldn't the source of momentum decay with height? if this is the case, then this should be rephrased as a modified wall model.*

*c. The references provided in Lines 41-43 suggest waves modify the wind profile through changes in surface stresses, not through injections of momentum across the surface layer: AlSam et al. (2015) and Yang et al. (2014) study how swell can modify wake propagation. Jenkins et al. (2012) use a coupled atmosphere-ocean model that modifies the wind field through changes in surface roughness. Kalviget al. (2014) resolve waves with a moving mesh, thus the wind profile is effectively modified by changes in surface roughness. Paskyabi et al. (2014) develop a wall model that accounts for wave-induced momentum fluxes. Porchetta et al. (2021) and Wu et al. (2020) use an atmosphere-wave coupled model, where the winds are modified by waves through changes in surface roughness. Zou et al. (2018) also focuses on a wall*

*model.*

**Response:** This is a good comment. We agree that waves modify the wind profile through changes in surface stress rather than through momentum injection through the surface layer. Waves can change the roughness length of the atmospheric subsurface, which in turn affects the momentum transport from the atmosphere to the ocean and to the waves. In this study, we argue that the coupled model does not account for the significant changes in roughness caused by large floating platforms affecting waves. This implies that the estimation of momentum fluxes in the sub-grid is incorrect, i.e. the momentum transport from the atmosphere to the ocean and waves needs to be reassessed. We believe that the loss of kinetic energy in the grid is not only due to the turbines, but also to changes in kinetic energy due to unresolved wind stress (friction) work. Thus, this is indeed a physical source of momentum. In the new wind farm parameterization scheme, this source really does not decrease with height. This is because momentum fluxes, heat fluxes, vary less with height in the near-surface layer (constant flux layer). Since the maximum height of the near-surface layer is about 100 meters, we also considered that the new scheme is only applicable up to a height of 100 meters (Figure R2).

[Figure]

Figure R2. Composition of the atmospheric boundary layer.

*Comment 3: Model configuration in Section 5.2: The authors use a 12 km horizontal grid spacing for their simulations. However, Tomaszewski and Lundquist (2020) show such coarse grid scan produce unrealistic impacts over a very broad region. Please explain your choice of grid spacing.*

**Response:** Thank you for your suggestion. The horizontal resolution of the previous WRF was really coarse, so we conducted new experiments. Two nested domains are used in WRF with their respective grid spacings of 9 and 3 km.

*Comment 4: Section 5.4: The authors conclude that Fitch overestimates wake effects. However, the FWFP is artificially accelerating wake recovery downstream of the*

*turbines. Thus, it is expected to have higher power production estimates and lower wake deficits in the FWFP*

*Lines 243-244: Adding a source to the momentum tendency is expected to accelerate wake recovery downstream of the turbines. Thus, is it reasonable to say that that Fitch underestimates power output? Rather, the momentum source in the FWFP accelerates wake recovery; thus, momentum availability increases amplifying power production.*

*Lines 257-258: same as above.*

**Response:** This is a good comment. We agree that the statement "Fitch underestimates power output" is not reasonable and have reworded it. We also agree that the additional momentum source in the FWFP affects wake recovery. Thus, the increased momentum availability increases power output. However, we do not believe that the power output increases for all turbines (Figure R3). We also wrote in the last paragraph of the paper that the decrease in significant wave height does not necessarily lead to higher wind speeds in the near-surface layer. This is because the associated iterative algorithms in WRF are too complex, so reducing the significant wave height also has the potential to reduce near-surface wind speeds. Figure R4 shows that the roughness length in the Taylor and Yelland scheme is determined by the significant wave height, the peak wave length, and the frictional velocity. Figure R5 shows that the roughness length in turn is involved in the calculation of the new friction velocity. The new friction velocity then determines the heat flux, moisture flux, etc., and is looped into the calculation of the next roughness length. Thus, it cannot simply be assumed that the additional momentum source in the FWFP increases momentum availability.

[Figure]

Figure R3. Power output differences between WRF-Fitch and WRF-FWFP cases.

Figure R4. Taylor and Yelland expression in WRF.

Figure R5. Code in WRF related to the calculation of frictional velocity.

*Minor comments:*

*1. I recommend English language revisions throughout the manuscript.*

**Response:** Thank you for your suggestion. We have revised the entire manuscript.

*2. Lines 22-24: What about coupled meso-microscale simulations? Coupled mesoscale-LES simulations using WRF can capture these effects, however, at a higher computational cost.*

**Response:** Thank you for your suggestion. A brief overview of the coupled mesoscale-LES simulations is given in lines 25-27.

*3. Line 31: Please add punctuation as: "... sink on the mean flow. Most of ..."*

**Response:** Thank you for your suggestion. We made modifications.

*4. Lines 27-44: I recommend splitting paragraph #2 in the introduction, perhaps at line 35.*

**Response:** Thank you for your suggestion. We have split the second paragraph into two parts.

*5. Lines 43-44: I would argue that the current parameterization can be suitable for floating offshore wind farms. Rather, the atmosphere-only model in WRF does not*

*capture changes in roughness length over the ocean caused by the presence of floating turbines*

**Response:** This is a good comment. We agree that atmosphere-only model cannot be used when applied to offshore wind farms. However, we believe that only semi-submersible floating wind turbines with large floating platforms have the greatest impact on local waves (Figure R6). Current wind farm parameterization schemes are not applicable to such floating wind turbines.

[Figure]

Figure R6. Floating wind turbine classification.

*6. Line 74: Please explain why you chose d = 20 m.*

**Response:** Thank you for your suggestion. This is a common draft depth for semi-submersible floating wind turbines.

*7. Captions should fully describe the figure. Please include additional information in all captions to make each figure self-explanatory. For example, include description of the different terms and symbols used in Figure 1, as well as the significance of the red contours.*

**Response:** Thank you for your suggestion. We have modified all captions to include detailed information.

*8. Figure 3: It is difficult to read the information within the grey area. Please use colors with higher contrast. Also, what is the meaning of the blue curves (presumably schematic for waves) to the side of the plot?*

**Response:** Thank you for your suggestion. We have redrawn the figure. The blue curve on the right side of the plot represents waves.

*9. Line 173: "The important point in the derivation ..." implies that the source of TKE in the Fitch parameterization is not important. Please rephrase.*

**Response:** Thank you for your suggestion. We have revised this sentence.

*10. Lines 193-201 and Figure 6: Please maintain consistency in your nomenclature (e.g., the authors use $u_{*,wt}$ in Eq. 17, but ustwt in Figure 6)*

**Response:** Thank you for your suggestion. We have redrawn the figure.

*11. Figures 11, 13, 14: It would be helpful to show the top and bottom of the turbine rotor layer for reference.*

**Response:** Thank you for your suggestion. We have redrawn the relevant figures. The diagram shows the hub height of the turbine and the top and bottom of the rotor with horizontal solid and dashed lines.

We thank you again for giving us an opportunity to revise this manuscript, and look forward to hearing from you.

Sincerely,

Shengli Chen

---

## Referee Report (RR1)

I appreciate the authors addressing virtually all of my comments satisfactorily, and greatly increasing the clarity and readability of the manuscript. And it remains an interesting application and one of potential great importance.

Unfortunately I am still troubled by my Comment 8 on old Line 184, which I noted at the time was my biggest concern. (And Reviewer 2's Comment 2 suggests they have a similar concern.) Specifically, the inclusion of an elevated momentum source due to increased friction velocity from the turbines. I think there is some general confusion on these matters, so some leeway can be permitted, but I would make the following points.

*) I am still not sure how new equation (22) is derived, specifically the derivation of the momentum flux term (delta_tau S Vijk (zk+1 - zk) in the mean kinetic energy budget, where delta_tau is the change in rho * ustar * ustar induced by turbine-impacted waves. Momentum flux times mean velocity is not a tendency of kinetic energy. (Negative) vertical gradient of momentum flux times mean velocity can be a tendency of mean kinetic energy. If the momentum flux really had this (constant with height) value up to 100 m and then became zero, you would have infinite accelerations at 100 m, and no accelerations above and below.

*) In reality, vertical turbulent momentum mixing in WRF is done with a turbulence closure parameterization, usually a function of the TKE equation and some diagnostics. Momentum flux convergence / divergence is one of these terms. The only role of surface roughness / momentum stress in this framework is as a lower boundary condition such that these divergences can be computed.

*) I would also say that there is not generally a 'constant momentum flux layer' in the atmosphere -- it is more correct to say that the surface layer is defined as a layer thin enough such that the momentum flux may be treated as approximately constant. Since it generally decreases on the scale of the boundary layer height, this is generally 10% of that value. But this means that wave impacts on surface roughness can cause drag to be transmitted throughout the depth of the PBL given enough turbulent transport.

*) Less essentially, it should be noted that 100 m can be well above the surface layer in marine conditions, in which case a constant stress formulation would be inappropriate.

*) In summary, without working through all details, I would advocate for wave-roughness modifications on momentum flux to be applied at the surface only, and then either transmitted throughout the PBL by the PBL parameterization, or perhaps somehow specified to decrease with height to the boundary layer top as long as double counting is avoided.

Minor comments:

Line 276: You say that 'the model is run for an additional 150 hours for further validation (0000 UTC 01 January to 0000 UTC 07 January)' but 00 UTC 01 Jan to 00 UTC 07 Jan is six days = 144 hours, and the six hour period for all later analyses falls within this interval. So I am still a little confused about the time intervals involved.

Response to comment 12, line 351: I could quibble about the degree that the Taylor Yelland parameterization is iterative. While this is technically true, the iterative aspect of the scheme is in a viscous term that is actually derived from laminar theory, and in practice provides a roughness length in conditions for which wave roughness is zero. In regions of interest for wave roughness, this term should hopefully have negligible effect, in which case iteration is not needed to determine the roughness height.

However, it is certainly true that the similarity schemes and turbulence physics of WRF overall are complex, and there are a number of reasons why reductions of SWH might not increase near-surface wind speeds. What I might suggest is to just remove the reference to '(Taylor and Yelland 2001)' here but leave it in Table 1.

---

## Referee Report (RR2)

**Review: A parameterization scheme for the floating wind farm in a coupled atmosphere-wave model (COAWST v3.7)**

I appreciate all the author's comments and changes to the manuscript. The manuscript improved considerably, but I believe there are still a couple of points that need to be addressed more thoroughly.

**Major Comments:**

1. Modifications to momentum equation:
    a. Momentum source: In the response to reviewers' comments, the authors argue that sub-grid momentum fluxes may be misrepresented when modeling floating wind turbines in mesoscale models. This is a valid and interesting hypothesis. However, the authors do not provide evidence to support this statement. More important, the authors do not provide evidence/references in their manuscript to justify adding a source of momentum and an explanation of why this source of momentum is added to the lowest 100 m. Making such a statement without referencing other work that highlights this problem would require either observational or high-fidelity simulation results. This is a crucial part of this manuscript that needs to be addressed prior to publication as it has first-order effects on wake recovery and, thus, on the power output of the model.
    b. Depth of momentum source: In the response to reviewers' comments, the authors argue that they add the source of momentum in the lowest 100 m of the atmosphere because this is the depth of the surface layer (constant flux later). However, the surface layer depth changes constantly, as the authors imply in Figure R22 where the depth of the surface layer is assumed to be between 50 and 100 m above the surface. For instance, for some stably stratified flows, the surface layer may be a couple of meters in depth and the boundary layer may be about 100 m in depth. Therefore, this assumption does not hold for simulating realistic atmospheric flows. Like I mentioned in the previous comment, making this assumption without referencing prior work would require observational evidence or results from high-fidelity simulations.

**Minor comments:**
1. It is not clear from Figure 10 that the FWFP simulations can produce lower power output compared with Fitch. From a visual inspection, Figure 10a does not have any clear red contours, which presumably mean Fitch produced more power than the FWFP. Also, the color bar on Figure 10b only has positive values. Why are there red contours in Figure R23, but there aren't any (at least not discernable) in Figure 10a?

---

## Referee Report (RR3)

**Review: A parameterization scheme for the floating wind farm in a coupled atmosphere-wave model (COAWST v3.7)**

I appreciate the authors response to all my comments and the changes made to the manuscript. I believe the changes made to the manuscript have made it clearer. I would appreciate clarification on a couple of points prior to publication.

1. Inflow wind speed in FWP: In Sect. 5.1, the authors propose an inflow wind speed to the turbine $V_{ijk|wt}$ that differs from the horizontal wind speed in the grid cell $V_{ijk}$. It is not clear how the inflow wind speed $V_{ijk|wt}$ is derived from $V_{ijk}$. The authors provide a short explanation on why $V_{ijk|wt}$ differs from $V_{ijk}$ in Lines 312-315. Perhaps a flow chart like Fig. 1 in DOI: 10.1175/MWR-D-20-0097.1 comparing the FWFP and Fitch WFP might help clarify this point.
2. Line 295-298: The authors suggest the turbines in the upstream region of the wind farm "absorb" less momentum in the FWFP than in the Fitch WFP. However, wouldn't the reduced friction velocity from the FWFP result in faster winds impacting the turbine? If so, then the turbines would extract more momentum from the wind in the FWFP compared to the Fitch WFP. Is it possible that the increased drag of the entire wind farm (increased power extraction) in the FWFP compared to the Fitch WFP produces a stronger blockage effect upstream of the wind farm?

---

## Author Response (AR2)

*I appreciate the authors addressing virtually all of my comments satisfactorily, and greatly increasing the clarity and readability of the manuscript. And it remains an interesting application and one of potential great importance.*

*Unfortunately I am still troubled by my Comment 8 on old Line 184, which I noted at the time was my biggest concern. (And Reviewer 2's Comment 2 suggests they have a similar concern.) Specifically, the inclusion of an elevated momentum source due to increased friction velocity from the turbines. I think there is some general confusion on these matters, so some leeway can be permitted, but I would make the following points.*

**Responses to the comments of Reviewer #1:**

**We sincerely thank the reviewer for the suggestions and comments that help us improve the quality of our manuscripts.**

*\*) I am still not sure how new equation (22) is derived, specifically the derivation of the momentum flux term (delta_tau S Vijk (zk+1 - zk) in the mean kinetic energy budget, where delta_tau is the change in rho \* ustar \* ustar induced by turbine-impacted waves. Momentum flux times mean velocity is not a tendency of kinetic energy. (Negative) vertical gradient of momentum flux times mean velocity can be a tendency of mean kinetic energy. If the momentum flux really had this (constant with height) value up to 100 m and then became zero, you would have infinite accelerations at 100 m, and no accelerations above and below.*

*\*) In reality, vertical turbulent momentum mixing in WRF is done with a turbulence closure parameterization, usually a function of the TKE equation and some diagnostics. Momentum flux convergence / divergence is one of these terms. The only role of surface roughness / momentum stress in this framework is as a lower boundary condition such that these divergences can be computed.*

*\*) I would also say that there is not generally a 'constant momentum flux layer' in the atmosphere -- it is more correct to say that the surface layer is defined as a layer thin enough such that the momentum flux may be treated as approximately constant. Since it generally decreases on the scale of the boundary layer height, this is generally 10% of that value. But this means that wave impacts on surface roughness can cause drag to be transmitted throughout the depth of the PBL given enough turbulent transport.*

*\*) Less essentially, it should be noted that 100 m can be well above the surface layer in marine conditions, in which case a constant stress formulation would be inappropriate.*

*\*) In summary, without working through all details, I would advocate for wave-roughness modifications on momentum flux to be applied at the surface only, and then either transmitted throughout the PBL by the PBL parameterization, or perhaps somehow specified to decrease with height to the boundary layer top as long as double counting is avoided.*

**Response:** Thank you for your suggestion. We agree that the previous equations are

not appropriate for the new wind farm parameterization scheme. In particular, we simply define the near-surface layer as a height of 100 meters above the sea surface, which should not be a constant. Following your suggestion that the roughness length modifications on momentum flux should be applied at the surface only, and then transmitted throughout the PBL by the PBL parameterization. We have modified the parameterization scheme with the following flow chart (Figure R11), the 3D wind speed at the wind turbine location is calculated by the PBL and passed to the wind farm parameterization to modify the inflow wind speed. We have re-run the simulations and made comparisons, and the new results are presented in the revised manuscript.

[Figure]

**Figure R11.** Flow chart of floating offshore wind farms parameterization implemented in the coupled model (HWAVE = significant wave height, LWAVEP = peak wave period, PWAVE = peak wave length, DEPTH= water depth, U10 = zonal wind at 10 m, V10 = meridional wind at 10 m, UST=  frictional velocity, USTWT= frictional velocity at the wind turbine, ZNT = roughness length, ZNTWT = roughness length at the wind turbine,    U3D = three-dimensional zonal winds, V3D = three-dimensional meridional winds, U3DWT = three-dimensional zonal winds at the wind turbine, V3DWT = three-dimensional meridional winds at the wind turbine, TKE = turbulent kinetic energy, du = zonal momentum increment, dv = meridional momentum increment).

We have re-run the simulation and made comparisons. A brief description of how the new results are different is given in this response, and the details are in the revised manuscript. The results indicate that the power output of the entire floating wind farm in the Fitch scheme is higher only (<3 %) in the upwind grid compared to the FWFP scheme. The power output is low in most of the other grids, within 20 % (Figure R12). There is also a significant difference in the wind speed deficit caused by the two schemes. As a result of the FWFP modification of the inflow wind speed, the wind speed in the upwind region increases (<0.4 m/s) relative to the Fitch scheme. Wind speeds in the downwind region are reduced, but to a greater extent, to within 1.8 m/s (Figure R13). The distribution of the differences in TKE corresponds well to the distribution of the differences in wind speed. Compared to the Fitch scheme, the

FWFP scheme generates less TKE in the upwind region (< 0.6 m2/s-2) and more TKE in the downwind region (< 1.4 m2/s-2) (Figure R14).

[Figure]

**Figure R12.** (a) (b) Power output of the WRF-Fitch case minus the WRF-FWFP case, but only positive values are shown in Figure R12b. The black dashed line indicates the outer boundary of the wind farm, and the black arrow indicates the wind direction. (c) Histogram of the relative difference between the power output of the WRF-Fitch case and the power output of the WRF-FWFP case. (d) Boxplot of the relative difference in the power output, the same as Figure 6.

[Figure]

**Figure R13**. Horizontal wind speed of (a) the WRF-FWFP case minus the WRF-CTL case and (b) the WRF-Fitch case minus the WRF-FWFP case at the hub height level. The red solid line indicates the outer boundary of the wind farm, and the green solid line indicates a cross section analyzed further.

[Figure]

**Figure R14**. Horizontal distribution of TKE differences at the top of the turbine between (a) WRF-FWFP and WRF-CTL cases and (b) WRF-Fitch and WRF-FWFP cases, near the sea surface between (c) WRF-FWFP and WRF-CTL cases and (d) WRF-Fitch and WRF-FWFP cases. The red solid line shows the outer boundary of the wind farm, and the pink solid line indicates a cross section analyzed further.

*Minor comments:*

*Line 276: You say that 'the model is run for an additional 150 hours for further validation (0000 UTC 01 January to 0000 UTC 07 January)' but 00 UTC 01 Jan to 00 UTC 07 Jan is six days = 144 hours, and the six hour period for all later analyses falls within this interval. So I am still a little confused about the time intervals involved.*

**Response:** Thank you for your suggestion. Sorry for the confusion, we modified it in line 279 of the revised manuscript. The following figure may help to understand this better. The simulations are integrated first for 12 hours without the turbines to reach a steady state, and then run for another 6 hours for comparison. The model is also run for an additional 126 hours for further validation.

[Figure]

*Response to comment 12, line 351: I could quibble about the degree that the Taylor Yelland parameterization is iterative. While this is technically true, the iterative*

*aspect of the scheme is in a viscous term that is actually derived from laminar theory, and in practice provides a roughness length in conditions for which wave roughness is zero. In regions of interest for wave roughness, this term should hopefully have negligible effect, in which case iteration is not needed to determine the roughness height.*

*However, it is certainly true that the similarity schemes and turbulence physics of WRF overall are complex, and there are a number of reasons why reductions of SWH might not increase near-surface wind speeds. What I might suggest is to just remove the reference to '(Taylor and Yelland 2001)' here but leave it in Table 1.*

**Response:** Thank you for your suggestion. We agree that the similarity schemes and turbulence physics of WRF are complex. Reduction of SWH impacts frictional velocity, roughness length, etc., which doesn't necessarily lead to an increase in near-surface wind speeds. And only in Table 1 have we retained '(Taylor and Yelland, 2001)'.

Thank you again for your great comment.

*I appreciate all the author's comments and changes to the manuscript. The manuscript improved considerably, but I believe there are still a couple of points that need to be addressed more thoroughly.*

**Responses to the comments of Reviewer #2:**

**We sincerely thank the reviewer for the suggestions and comments that help us improve the quality of our manuscripts.**

**Major Comments:**

*1. Modifications to momentum equation:*
*a. Momentum source: In the response to reviewers' comments, the authors argue that sub-grid momentum fluxes may be misrepresented when modeling floating wind turbines in mesoscale models. This is a valid and interesting hypothesis. However, the authors do not provide evidence to support this statement. More important, the authors do not provide evidence/references in their manuscript to justify adding a source of momentum and an explanation of why this source of momentum is added to the lowest 100 m. Making such a statement without referencing other work that highlights this problem would require either observational or high-fidelity simulation results. This is a crucial part of this manuscript that needs to be addressed prior to publication as it has first-order effects on wake recovery and, thus, on the power output of the model.*

*b. Depth of momentum source: In the response to reviewers' comments, the authors argue that they add the source of momentum in the lowest 100 m of the atmosphere because this is the depth of the surface layer (constant flux later). However, the surface layer depth changes constantly, as the authors imply in Figure R22 where the depth of the surface layer is assumed to be between 50 and 100 m above the surface. For instance, for some stably stratified flows, the surface layer may be a couple of meters in depth and the boundary layer may be about 100 m in depth. Therefore, this assumption does not hold for simulating realistic atmospheric flows. Like I mentioned in the previous comment, making this assumption without referencing prior work would require observational evidence or results from high-fidelity simulations.*

**Response:** Thank you for your suggestion. We agree that it is not always reasonable to add additional sources of momentum directly to the wind farm parameterization, not to mention that it is indeed difficult to define whether they are positive or negative. We also agree that the thickness of the near-surface layer cannot simply be defined as a constant (100 m). Especially when the subsurface is marine, the thickness is usually smaller. We have revised the new wind farm parameterization, taking into account the suggestions you made with another reviewer. Please refer to the flow chart below (Figure R21). We used the Planetary Boundary Layer Driver module in WRF to calculate the 3D wind speed as the frictional velocity, roughness length, etc. are changed. This 3D wind speed is passed to the wind farm parameterization as the new inflow wind speed.

[Figure]

**Figure R21.** Flow chart of floating offshore wind farms parameterization implemented in the coupled model (HWAVE = significant wave height, LWAVEP = peak wave period, PWAVE = peak wave length, DEPTH= water depth, U10 = zonal wind at 10 m, V10 = meridional wind at 10 m, UST= frictional velocity, USTWT= frictional velocity at the wind turbine, ZNT = roughness length, ZNTWT = roughness length at the wind turbine, U3D = three-dimensional zonal winds, V3D = three-dimensional meridional winds, U3DWT = three-dimensional zonal winds at the wind turbine, V3DWT = three-dimensional meridional winds at the wind turbine, TKE = turbulent kinetic energy, du = zonal momentum increment, dv = meridional momentum increment).

We have re-run the simulation and made comparisons. A brief description of how the new results are different is given in this response, and the details are in the revised manuscript. The results indicate that the power output of the entire floating wind farm in the Fitch scheme is higher only (<3 %) in the upwind grid compared to the FWFP scheme. The power output is low in most of the other grids, within 20 % (Figure R22). There is also a significant difference in the wind speed deficit caused by the two schemes. As a result of the FWFP modification of the inflow wind speed, the wind speed in the upwind region increases (<0.4 m/s) relative to the Fitch scheme. Wind speeds in the downwind region are reduced, but to a greater extent, to within 1.8 m/s (Figure R23). The distribution of the differences in TKE corresponds well to the distribution of the differences in wind speed. Compared to the Fitch scheme, the FWFP scheme generates less TKE in the upwind region (< 0.6 m2/s-2) and more TKE in the downwind region (< 1.4 m2/s-2) (Figure R24).

[Figure]

**Figure R22.** (a) (b) Power output of the WRF-Fitch case minus the WRF-FWFP case, but only positive values are shown in Figure R12b. The black dashed line indicates the outer boundary of the wind farm, and the black arrow indicates the wind direction. (c) Histogram of the relative difference between the power output of the WRF-Fitch case and the power output of the WRF-FWFP case. (d) Boxplot of the relative difference in the power output, the same as Figure 6.

[Figure]

**Figure R23.** Horizontal wind speed of (a) the WRF-FWFP case minus the WRF-CTL case and (b) the WRF-Fitch case minus the WRF-FWFP case at the hub height level. The red solid line indicates the outer boundary of the wind farm, and the green solid line indicates a cross section analyzed further.

[Figure]

**Figure R24**. Horizontal distribution of TKE differences at the top of the turbine between (a) WRF-FWFP and WRF-CTL cases and (b) WRF-Fitch and WRF-FWFP cases, near the sea surface between (c) WRF-FWFP and WRF-CTL cases and (d) WRF-Fitch and WRF-FWFP cases. The red solid line shows the outer boundary of the wind farm, and the pink solid line indicates a cross section analyzed further.

**Minor Comments:**

*1. It is not clear from Figure 10 that the FWFP simulations can produce lower power output compared with Fitch. From a visual inspection, Figure 10a does not have any clear red contours, which presumably mean Fitch produced more power than the FWFP. Also, the color bar on Figure 10b only has positive values. Why are there red contours in Figure R23, but there aren't any (at least not discernable) in Figure 10a?*

**Response:** This is a good comment. The previous Figure 10 and Figure R23 are actually the same, only the range of the color bars is different. But we have performed new experiments.

Thank you again for your great comment.

---

## Author Response (AR3)

*The authors have greatly improved the FWFP formulation and the manuscript as a whole. I recommend publication after minor revisions. I appreciate the authors response to all my comments and the changes made to the manuscript. I believe the changes made to the manuscript have made it clearer. I would appreciate clarification on a couple of points prior to publication.*

**Responses to the comments of Reviewer #1:**

**We sincerely thank the reviewer for the suggestions and comments that help us improve the quality of our manuscripts.**

*1. Inflow wind speed in FWP: in Sect. 5.1, the authors propose an inflow wind speed to the turbine $V_{ijk|wt}$ that differs from the horizontal wind speed in the grid cell $V_{ijk}$. It is not clear how the inflow wind speed $V_{ijk|wt}$ is derived from $V_{ijk}$. The authors provide a short explanation on why $V_{ijk|wt}$ differs from $V_{ijk}$ in Lines 312-315. Perhaps a flow chart like Fig. 1 in DOI: 10.1175/MWR-D-20-0097.1 comparing the FWFP and Fitch WFP might help clarify this point.*

**Response:** That's a good comment. A more detailed explanation is needed to give the reader an idea of what factors influence the change in the inflow wind speed. The largest uncertainty comes from the surface parameterization scheme (module_sf_mynn.F) in the WRF model. The attenuation of the effective wave height by the floating platform leads to changes in a number of variables that together determine the (inflow) wind speed (Figure 1). We have added this in lines 312-315 of the revised manuscript.

[Figure]

**Figure 1**. Flowchart of the computation of surface layer variables in the WRF model.

*2. Line 295-298: The authors suggest the turbines in the upstream region of the wind farm "absorb" less momentum in the FWFP than in the Fitch WFP. However wouldn't the reduced friction velocity from the FWFP result in faster winds impacting the turbine? If so, then the turbines would extract more momentum from the wind in the FWFP compared to the Fitch WFP. is it possible that the increased drag of the entire*

*wind farm (increased power extraction) in the FWFP compared to the Fitch WFP produces a stronger blockage effect upstream of the wind farm?*

**Response:** Thank you for your suggestion. It is true that the "the upstream region of the wind farm absorbs less momentum in the FWFP than in the Fitch WFP" is not quite correct. We have made modifications in lines 294-295 of the revised manuscript. 93.2% of the turbines have a lower power prediction for the Fitch scheme than for the FWFP scheme. In fact, it is very likely that this is because the FWFP takes into account that the frictional velocity is lower (the inflow wind speed is higher). The results for the upstream turbines are likely due to stronger blockage effects, but the area affected is small.

We thank you again for giving us an opportunity to revise this manuscript, and look forward to hearing from you.

*I appreciate the authors revising their scheme to take into account my concerns and redoing the experiments. The new results seem reasonable, and I think the manuscript is almost there. However, I just have a few hopefully minor concerns:*

**Responses to the comments of Reviewer #2:**

**We sincerely thank the reviewer for the suggestions and comments that help us improve the quality of our manuscripts.**

*1) Section 5.1: A little more explanation of these equations / references might be good. What is C_P in equation 26? What is '|wt'? Isn't (23) just a rearrangement of (22)? Can you explicitly state what the change relative to Fitch is? (I think just V|wt which comes from machine learning surface layer with SWAN inputs?)*

**Response:** Thank you for your suggestion. In section 5.1, we add relevant content. $C_P$ in Equation (26) is the power coefficient. $V_{ijk|wt}$ is the recalculated inflow wind speed at the wind turbine site. Equation (23) is actually a rearrangement of equation (22), which makes it clearer to modify the momentum tendency term in the wind farm parameterization. In the Fitch wind farm parameterization (module_wind_fitch.F), only the inflow wind speed ($V_{ijk|wt}$) is changed.

*2) Figure 13 caption, lines 305-315, Figure 16 caption: when you mention differences and their sign, please indicate the order of subtraction of the two terms.*

**Response:** Thank you for your suggestion. We have modified the caption for these two figures.

*3) Figure 12, 14, 15: please indicate in the cross sections which end is 'upstream'.*

**Response:** Thank you for your suggestion. We have made changes to the three figures.

*4) Conclusion: When you discuss the relative differences between Fitch and FWFP, I would first mention the downstream differences, because in general those are substantially larger than the upstream differences.*

**Response:** That's a good comment. We have made modifications in lines 413-414 of the revised manuscript. We first described the downstream differences.

*5) Conclusion, throughout: if you can provide it, it would be nice to have a concise statement of how the expected impact of floating wind turbines on waves (I think, a reduction of SWH and roughness), which is the prime innovation of your scheme, leads to your modeled sensitivities on power production (more power than in default scheme, along with less wind speeds downstream).*

**Response:** That's a good comment. We have added a corresponding statement in lines 415-416 of the revised manuscript. 93.2% of the turbines have a lower power prediction for the Fitch scheme than for the FWFP scheme. In fact, it is very likely that this is because the FWFP takes into account that the frictional velocity is lower (the inflow wind speed is higher).

*6) Last sentence of conclusion: 'In order to better evaluate the power output of floating wind farms and their impacts on the environment, it is necessary to improve the offshore wind farms parameterization.' Can you give just a few reasons why you conclude this?*

**Response:** That's a good comment. The difference in power output between fixed (Fitch scheme) and floating (FWFP scheme) offshore wind farms is already apparent. Taking into account the attenuation of the significant wave height by the floating platform, the difference in power output can reach 12% on average. The difference in environmental impact is likely to be much smaller. We decided to remove this sentence because it could be misleading.

*7) Though the grammar and clarity are improved, a further technical edit could be used.*

**Response:** Thank you for your suggestion. We have carefully checked the whole manuscript and made necessary technical corrections.

Thank you again for your great comment.